# FAIRNESS-AWARE GRAPH LEARNING: A BENCHMARK

## ABSTRACT

Fairness-aware graph learning has gained increasing attention in recent years. Nevertheless, there lacks a comprehensive benchmark to evaluate and compare different fairness-aware graph learning methods, which blocks practitioners from choosing appropriate ones for broader real-world applications. In this paper, we present an extensive benchmark on ten representative fairness-aware graph learning methods. Specifically, we design a systematic evaluation protocol and conduct experiments on seven real-world datasets to evaluate these methods from multiple perspectives, including group fairness, individual fairness, the balance between different fairness criteria, and computational efficiency. Our in-depth analysis reveals key insights into the strengths and limitations of existing methods. Additionally, we provide practical guidance for applying fairness-aware graph learning methods in applications. To the best of our knowledge, this work serves as an initial step towards comprehensively understanding representative fairness-aware graph learning methods to facilitate future advancements in this area.

## 1 INTRODUCTION

Graph-structured data has become ubiquitous across a plethora of real-world applications (Hu et al., 2020; Ying et al., 2019; Dong et al., 2023a; Narayanan et al., 2017), such as social network analysis (Cho et al., 2011; Leskovec et al., 2010; Leskovec & Mcauley, 2012), biological network modeling (Zitnik et al., 2018; Pavlopoulos et al., 2011; Zitnik & Leskovec, 2017), and traffic pattern prediction (Yuan & Li, 2021; Atluri et al., 2018; Derrow-Pinion et al., 2021). To gain a deeper understanding of graph-structured data, graph learning methods, such as Graph Neural Networks (GNNs), are emerging as widely adopted and versatile methods to handle predictive tasks on graphs (Wu et al., 2020; Zhou et al., 2020; Wu et al., 2022; You et al., 2019). However, as we aim for improving utility (e.g., accuracy in node classification tasks), existing graph learning methods have also been found to constantly exhibit algorithmic bias in recent studies, which has raised significant societal concern and attracted attention from both industry and academia (Dong et al., 2023b; Choudhary et al., 2022; Wu et al., 2021). For example, financial agencies have been relying on GNNs to perform decision making in financial services (Wang et al., 2021; Song et al., 2023), e.g., determining whether each loan application should be approved or not based on transaction networks of bank clients. Nevertheless, the outcomes have been found to exhibit bias, such as racial disparities in the rejection rate (Song et al., 2023). As a consequence, addressing the fairness concerns for graph learning methods has become an urgent need (Dong et al., 2023b; Dai et al., 2022), especially under high-stake real-world applications such as financial lending (Song et al., 2023; Li et al., 2020) and healthcare decision making (Dai et al., 2022; Anderson & Visweswaran, 2024).

In recent years, various techniques, such as adversarial training (Dai & Wang, 2021; Jiang et al., 2024; Ling et al., 2023; Cong et al., 2023), optimization regularization (Agarwal et al., 2021; Jiang et al., 2022; Rahmattalabi et al., 2019), and graph structure learning (Dong et al., 2022; Zhang et al., 2024; Zhang & Ramesh, 2020), have been adopted to address the fairness concerns in graph learning methods. Nevertheless, despite these existing efforts, we have not yet seen extensive deployment of these fairness-aware graph learning methods. A primary obstacle lies in the lack of a comprehensive comparison across existing fairness-aware graph learning methods, which makes it difficult for practitioners to choose the appropriate ones to use. In fact, a comprehensive comparison of existing fairness-aware graph learning methods not only tells the best-in-class methods under different settings (e.g., different evaluation metrics and datasets from different domains) but also provides a guideline for practitioners to understand the strengths and limitations of different methods in multiple aspects,

such as utility, fairness, and efficiency. As such, comprehensively comparing the performances between different graph learning methods becomes an urgent need to facilitate a broader application of fairness-aware graph learning methods.

Multiple existing works have explored to compare different fairness-aware graph learning methods. For example, Chen et al. (Chen et al., 2024) proposed to categorize and compare existing fairness-aware GNNs by their input, main techniques, and tasks. However, the overwhelming focus on GNNs narrows down the scope of comparison. Another study from Laclau et al. (Choudhary et al., 2022) delivers a more comprehensive comparison of graph learning methods. However, it did not involve any quantitative performance comparison, which thus jeopardizes its practical value for practitioners. In fact, it is challenging to provide a quantitative performance comparison on fairness-aware graph learning methods due to their inconsistencies in terms of the studied fairness notions, experimental settings, and learning tasks. Therefore, lacking quantitative performance comparison becomes a common flaw for most of the related studies (Dai et al., 2022). More recently, Qian et al. (Qian et al., 2024) took an early step to present a quantitative performance benchmark in the area of graph learning. However, they only focus on the comparison of two fairness-aware GNNs, which thus blocks a broader understanding in a broader area of graph learning. Therefore, comprehensive performance comparison of fairness-aware graph learning methods remains underexplored.

In this paper, we take an initial step to comprehensively evaluate the performance differences between the most representative fairness-aware graph learning methods. Specifically, we first design a systematic evaluation protocol, which helps ensure consistent settings for the evaluation of different graph learning methods. Second, we collect ten of the most representative graph learning methods and present a comprehensive benchmark on seven real-world graph datasets (including five commonly used and two newly constructed ones) from different perspectives, such as different datasets, fairness notions, and evaluation metrics. Finally, we perform an in-depth analysis based on the experimental results and reveal key insights into the strengths and limitations associated with these fairness-aware graph learning methods. We also provide guidance for practitioners to choose appropriate ones to use, which further facilitates the practical significance of this study.

The main contributions of this paper are summarized as follows:

- **Experimental Protocol Design.** We design a systematic evaluation protocol, which enables the comparison between different fairness-aware graph learning methods under consistent settings. To the best of our knowledge, our work serves as the first step towards comprehensively evaluating the performance of fairness-aware graph learning methods.
- **Comprehensive Benchmark.** We conduct extensive experiments on seven real-world attributed graph datasets (including five commonly used and two newly constructed ones) and present a comprehensive benchmark over ten fairness-aware graph learning methods, which reveals key insights in understanding their strengths and limitations.
- **Multi-Perspective Analysis & Guidance.** We present four significant research questions and perform in-depth analysis from different perspectives based on the benchmarking results. Meanwhile, we also introduce a guide for practitioners to help them choose appropriate methods in real-world applications.

## 2 PRELIMINARIES

**Background.** We use $\mathcal{G} = \{\mathcal{V}, \mathcal{E}\}$ to denote a graph, where $\mathcal{V}$ denotes the set of $n$ nodes and $\mathcal{E}$ represents the set of edges. Here, each node is equipped with an attribute vector, which makes the graph an attributed graph. In this paper, we focus on node classification, which is among the most widely studied graph learning tasks. Typically, in node classification, a graph machine learning model can be represented as a function $f : (\mathcal{V}, \mathcal{E}) \to \hat{Y} \in \mathbb{R}^{n \times c}$, which maps each node $v \in \mathcal{V}$ into a $c$-dimensional matrix $\hat{Y}$. Each row in $\hat{Y}$ (denoted as $\hat{y}_i$ for the $i$-th row) is a vector indicating the predicted probability distribution across different classes, and $c$ denotes the total number of classes. Meanwhile, the matrix of ground truth labels $Y \in \{0, 1\}^{n \times c}$ is provided as the supervision for optimization. The primary goal of fairness-aware graph machine learning is to ensure $\hat{Y}$ bears high levels of utility and fairness at the same time. Without loss of generality, we conduct benchmarking experiments on the popular graph learning task of binary node classification (i.e., $c = 2$), which aligns with most works in this area (Dong et al., 2023b; Dai & Wang, 2021; Kose & Shen, 2022).

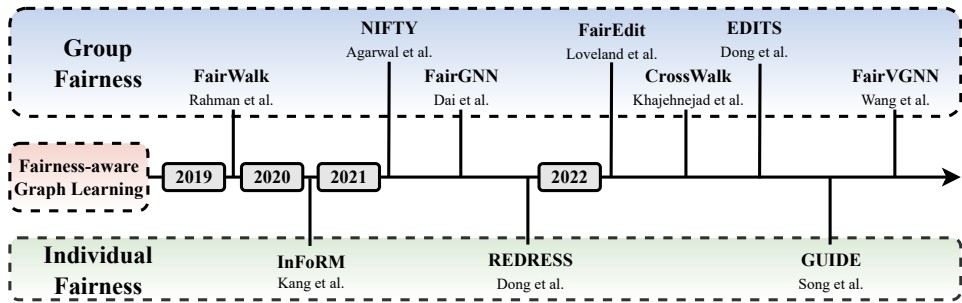

Figure 1: A timeline of the representative fairness-aware graph learning methods.

**Timeline of the Collected Graph Learning Models.** To provide a global understanding of fairness-aware graph learning methods, we present a high-level overview of the timeline of the representative explorations, which is shown in Figure 1. Specifically, we group these works by the fairness notions they focus on, including group fairness and individual fairness (Dong et al., 2023b). Group fairness emphasizes that the graph learning methods should not yield discriminatory predictions against any demographic subgroups (Dong et al., 2023b; Hardt et al., 2016), where the subgroups are determined by certain categorical sensitive attributes such as gender or race (Mehrabi et al., 2021; Dwork et al., 2012). On the other hand, individual fairness argues that similar individuals should be treated similarly (Dwork et al., 2012), i.e., the outcomes corresponding to a pair of individuals in the output space should be close if they are close in the input space (Dong et al., 2023b; Kang et al., 2020b).

**Notions and Metrics for Group Fairness.** Here, we present the representative notions and metrics under *Group Fairness*. **(1) Statistical Parity.** Statistical parity requires that the probability of yielding positive predictions should be the same across different demographic subgroups (Dong et al., 2023b; Dwork et al., 2012). Here, the rationale is that positive predictions correspond to beneficial decisions in a plethora of real-world applications (Hardt et al., 2016). A commonly used metric to quantify to what extent statistical parity is violated is $\Delta_{SP}$, which is given by

$$\Delta_{SP} = |P(\hat{Y} = 1 \mid S = 0) - P(\hat{Y} = 1 \mid S = 1)|, \tag{1}$$

where $\hat{Y}, S \in \{0, 1\}$ denote random variables for the predicted label and the sensitive attribute of any given individual, respectively. **(2) Equal Opportunity.** Equal opportunity requires that the probability of yielding positive predictions should be the same for those who have a positive ground truth across different demographic subgroups (Hardt et al., 2016). Different from statistical parity, equal opportunity aims to protect individuals' advantaged qualifications against bias arising from subgroup membership (Hardt et al., 2016). $\Delta_{EO}$ is commonly used to measure to what extent equal opportunity is violated, which is given by

$$\Delta_{EO} = |P(\hat{Y} = 1|Y = 1, S = 0) - P(\hat{Y} = 1|Y = 1, S = 1)|, \tag{2}$$

where $Y$ is the random variable of the ground truth for any given individual. **(3) Utility Difference-Based Fairness.** Its rationale is to reveal the largest utility gap between different demographic subgroups (Ali et al., 2021; Stoica et al., 2020; Rahmattalabi et al., 2021). A commonly used metric is the maximum utility difference across all pairs of demographic subgroups (denoted as $\Delta_{\text{Utility}}$). Here, utility refers to the performance in downstream node classification tasks (such as AUC-ROC score), and $\Delta_{\text{Utility}}$ serves as a fairness metric characterizing such performance gap between different demographic subgroups.

**Notions and Metrics for Individual Fairness.** We now present the representative notions and metrics under *Individual Fairness*. Different from group fairness, individual fairness does not rely on sensitive attributes. Instead, the rationale of individual fairness is to *treat similar individuals similarly* (Dwork et al., 2012). We introduce three notions and their corresponding metrics below. *(1) Lipschitz-Based Individual Fairness.* This notion argues that the (scaled) distance between individuals in the output space should be smaller or equal to their distance in the input space (Kang et al., 2020b). The level of the exhibited bias under this notion is measured by

$$B_{\text{Lipschitz}} = \sum_{i} \sum_{j, j \neq i} \|\hat{\boldsymbol{y}}_i - \hat{\boldsymbol{y}}_j\|_F \cdot \boldsymbol{S}_{ij}, \tag{3}$$

Table 1: Statistics of the collected real-world graph datasets.

| Dataset | Pokec-z | Pokec-n | German Credit | Credit Defaulter | Recidivism | AMiner-S | AMiner-L |
|---|---|---|---|---|---|---|---|
| **#Nodes** | 67,796 | 66,569 | 1,000 | 30,000 | 18,876 | 39,424 | 129,726 |
| **#Edges** | 882,765 | 729,129 | 24,970 | 200,526 | 403,977 | 52,460 | 591,039 |
| **#Attributes** | 276 | 265 | 27 | 13 | 18 | 5,694 | 5,694 |

where the $S$ is an oracle similarity matrix that describes the similarity between nodes in the input space. *(2) Ranking-Based Individual Fairness.* This notion requires that the rankings of the similarity between each individual and all other individuals should be the same between the input and output space (Dong et al., 2021b). The average top-$k$ similarity between the two ranking lists in the input and output spaces over all individuals is adopted as the fairness metric, where NDCG@$k$ is a common ranking similarity metric, which we denote as $B_{\text{ranking}}$. *(3) Ratio-Based Individual Fairness.* This notion requires that different demographic subgroups should bear similar levels of individual fairness (Song et al., 2022). *Group Disparity of Individual Fairness (GDIF)* is introduced as the metric, which is given by

$$GDIF = \sum_{i,j}^{1 \le i < j \le m} \max \left( \frac{B_{\text{Lipschitz}}^{(i)}}{B_{\text{Lipschitz}}^{(j)}}, \frac{B_{\text{Lipschitz}}^{(j)}}{B_{\text{Lipschitz}}^{(i)}} \right), \tag{4}$$

where $B_{\text{Lipschitz}}^{(i)}$ and $B_{\text{Lipschitz}}^{(j)}$ are the subgroup-level $B_{\text{Lipschitz}}$ from two demographic subgroups $i$ and $j$; $m$ is the total number of subgroups.

## 3 BENCHMARK DESIGN

In this section, we introduce the design of our benchmark. Specifically, we first present the experimental settings and implementation details of our benchmark. Then we introduce four main research questions we aim to explore in this paper. We note that our experiments are conducted based on node classification, since most commonly used fairness metrics are defined for classification.

### 3.1 EXPERIMENTAL SETTINGS AND IMPLEMENTATIONS

Here we introduce the experimental settings, including benchmark datasets, collected fairness-aware graph learning methods, and the implementation details regarding this newly introduced benchmark.

**Benchmark Datasets.** We collected seven real-world attributed graph datasets of different scales in this benchmark paper, including five existing commonly used ones and two newly constructed ones. These datasets include (1) *Pokec-z (Takac & Zabovsky, 2012)*: social network data; (2) *Pokec-n (Takac & Zabovsky, 2012)*: social network data; (3) *German Credit (Markelle Kelly)*: a graph based on financial credit; (4) *Credit Defaulter (Yeh & Lien, 2009)*: a graph over financial agency clients; (5) *Recidivism (Jordan & Freiburger, 2015)*: a graph over defendants; (6) *AMiner-S* (newly constructed): a co-authorship graph over researchers; *(5) AMiner-L* (newly constructed): a co-authorship graph over researchers. We present the statistics of the collected attributed graph datasets above in Table 1, and a more detailed dataset introduction is given in Appendix.

**Fairness-Aware Graph Learning Models.** We collect ten of the most representative graph learning methods for comparison. We provide a brief introduction for each of them below, where the fairness notion they focus on is marked out in brackets. (1) *FairWalk (group fairness).* FairWalk (Rahman et al., 2019) is a fairness-aware graph learning method based on DeepWalk, where it achieves bias mitigation by balancing the transition probabilities between different demographic subgroups. (2) *CrossWalk (group fairness).* CrossWalk (Khajehnejad et al., 2022) is a fairness-aware graph learning method. Specifically, it is developed based on DeepWalk, where such an algorithm achieves bias mitigation by steering random walks across demographic subgroup boundaries for representation learning. (3) *FairGNN (group fairness).* FairGNN (Dai & Wang, 2021) is a fairness-aware graph learning method base on GNNs, where it achieves bias mitigation by incorporating an adversary to wipe out the information of sensitive attributes in the learned node representations. (4) *NIFTY (group fairness).* NIFTY (Agarwal et al., 2021) is a fairness-aware graph learning method based on GNNs, where it achieves bias mitigation with an additional optimization regularization term based on counterfactual sensitive attribute perturbation. (5) *EDITS (group fairness).* EDITS (Dong et al., 2022)

is a fairness-aware graph learning framework designed in a pre-processing manner, where it achieves bias mitigation by minimizing the distribution difference between nodes from different demographic subgroups in the node attribute space. (6) *FairEdit (group fairness)*. FairEdit (Loveland et al., 2022) is a fairness-aware graph learning method based on GNNs, where it optimizes the performance on fairness by modifying the graph topology. (7) *FairVGNN (group fairness)*. FairVGNN (Wang et al., 2022) is a fairness-aware graph learning method based on GNNs, where it achieves bias mitigation by identifying and masking sensitive-correlated attribute dimensions. (8) *InFoRM (individual fairness)*. InFoRM (Kang et al., 2020b) is a fairness-aware graph learning method that can be adapted to different models, where it achieves bias mitigation by incorporating a fairness-aware optimization objective based on the Lipschitz condition. (9) *REDRESS (individual fairness)*. REDRESS (Dong et al., 2021b) is a fairness-aware graph learning method based on GNNs, where it proposes a fairness-aware optimization objective to improve performance on ranking-based fairness. (10) *GUIDE (individual fairness)*. GUIDE (Song et al., 2022) is a fairness-aware graph learning method based on GNNs, where it uses a fairness-aware optimization objective to enforce similar levels of Lipschitz-based individual fairness across different demographic subgroups.

**Implementation Details.** All benchmarking experiments are implemented with PyTorch and performed on an Nvidia A100 GPU. We obtain the best hyper-parameters by selecting the lowest loss values on the validation node set via grid search, and all results are reported with standard deviation from three different runs. For all GNNs, we adopt the most widely used GCN unless otherwise specified. Comprehensive experimental details, including open-source URLs of the algorithms we have used for reproducibility purposes, are introduced in Appendix.

## 3.2 RESEARCH QUESTIONS

**RQ 1: How well can those representative methods perform under group fairness?**

**Significance & Experimental Design.** Understanding the performance of graph learning methods in terms of group fairness is crucial since it addresses the bias that may arise in applications due to sensitive attributes such as race, gender, and age. We evaluate the collected methods focusing on group fairness on both utility and fairness. Here we adopt the AUC-ROC score as an exemplary metric for utility, while $\Delta_{\text{SP}}$, $\Delta_{\text{EO}}$, and $\Delta_{\text{Utility}}$ are utilized as the metrics for fairness (as in Section 2).

**RQ 2: How well can those representative methods perform under individual fairness?**

**Significance & Experimental Design.** Evaluating individual fairness helps to identify and reduce discriminatory practices at the individual level, which is more granular compared with group fairness. To answer this question, we evaluate the collected methods focusing on individual fairness from the perspective of both utility and fairness. Here, we adopt the AUC-ROC score for utility evaluation, while $B_{\text{Lipschitz}}$, NDCG@$k$, and GDIF are adopted as the metrics for fairness (as in Section 2).

**RQ 3: How well can existing methods balance different fairness criteria?**

**Significance & Experimental Design.** Understanding how graph learning methods balance different fairness criteria is vital when multiple criteria need to be considered simultaneously (Zhan et al., 2024; Sirohi et al., 2024; Dai et al., 2022). Considering the scarcity of methods under individual fairness, we focus on group fairness for this research question. Specifically, we measure the average ranking corresponding to these methods on $\Delta_{\text{SP}}$, $\Delta_{\text{EO}}$, and $\Delta_{\text{Utility}}$, where a lower average ranking indicates better performance.

**RQ 4: How well can those representative methods perform in terms of efficiency?**

**Significance & Experimental Design.** Ensuring that fairness-aware graph learning methods are computationally feasible is essential for their usability in real-world applications. To answer this question, we evaluate the collected methods by their utility vs. running time on each dataset. Better utility with less running time indicates better efficiency.

## 4 EMPIRICAL INVESTIGATION

In this section, we present benchmarking results and in-depth analysis to answer the four research questions in Section 3.2. Specifically, we first assess group fairness (RQ1) using metrics like statistical parity and equal opportunity, followed by individual fairness (RQ2), which ensures similar treatment

Table 2: Comparison of graph learning methods focusing on group fairness. Note that results include AUC-ROC score and $\Delta_{\text{SP}}$, and complete results are in Appendix. The best ones are in **bold**; the second best ones are underlined; OOM denotes out-of-memory.

| Metrics | Models | Pokec-z | Pokec-n | German Credit | Credit Defaulter | Recidivism | AMiner-S | AMiner-L |
|---|---|---|---|---|---|---|---|---|
| **AUC-ROC** | DeepWalk | 66.50 (± 1.34) | 61.85 (± 1.06) | 56.90 (± 1.75) | 53.61 (± 0.66) | 87.18 (± 1.34) | 73.58 (± 0.43) | 82.68 (± 3.28) |
| | FairWalk | 64.92 (± 0.43) | 61.52 (± 0.34) | 54.05 (± 0.83) | 55.51 (± 0.29) | 72.09 (± 0.11) | 65.35 (± 0.54) | 88.72 (± 0.08) |
| | CrossWalk | 58.99 (± 0.27) | 62.98 (± 0.27) | 51.42 (± 0.43) | 54.50 (± 0.42) | 82.89 (± 0.11) | 64.44 (± 0.52) | **89.67** (± 0.04) |
| | GNN | 64.16 (± 0.62) | 67.05 (± 1.14) | **67.36** (± 3.59) | 62.62 (± 0.51) | 84.60 (± 2.10) | 81.95 (± 1.46) | 86.82 (± 0.11) |
| | FairGNN | 69.47 (± 1.04) | 68.51 (± 0.51) | 52.91 (± 2.15) | 56.73 (± 3.16) | **92.87** (± 2.42) | **86.23** (± 0.14) | OOM |
| | NIFTY | 62.58 (± 0.14) | 66.78 (± 0.82) | 62.94 (± 5.78) | 61.85 (± 0.70) | 85.58 (± 0.83) | 79.28 (± 0.15) | 86.62 (± 0.69) |
| | EDITS | OOM | OOM | 60.02 (± 1.10) | 61.14 (± 0.36) | 92.34 (± 0.31) | OOM | OOM |
| | FairEdit | OOM | OOM | 56.30 (± 2.33) | 62.50 (± 0.61) | 81.97 (± 0.48) | OOM | OOM |
| | FairVGNN | **71.19** (± 0.94) | **70.14** (± 0.55) | 65.48 (± 3.46) | **68.81** (± 0.81) | 84.74 (± 2.70) | OOM | OOM |
| **$\Delta_{\text{SP}}$** | DeepWalk | 5.49 (± 1.07) | 5.90 (± 0.88) | 10.4 (± 1.01) | 6.69 (± 0.31) | 6.50 (± 0.18) | 6.75 (± 0.29) | 6.41 (± 0.46) |
| | FairWalk | **0.60** (± 1.89) | 0.29 (± 2.12) | 3.36 (± 1.01) | 6.20 (± 0.32) | **4.67** (± 0.33) | **3.06** (± 0.32) | **4.28** (± 0.17) |
| | CrossWalk | 1.75 (± 1.17) | **0.21** (± 1.63) | 0.35 (± 1.75) | 6.35 (± 0.51) | 5.14 (± 0.21) | 3.59 (± 0.43) | 5.60 (± 0.42) |
| | GNN | 10.4 (± 1.46) | 14.7 (± 0.40) | 32.4 (± 1.93) | 20.6 (± 4.34) | 8.54 (± 0.10) | 7.28 (± 0.31) | 6.75 (± 0.00) |
| | FairGNN | 2.06 (± 1.82) | 8.11 (± 1.16) | 14.2 (± 0.83) | 2.51 (± 5.61) | 7.48 (± 0.30) | 5.36 (± 0.27) | OOM |
| | NIFTY | 2.48 (± 0.47) | 2.42 (± 0.84) | 0.26 (± 0.41) | 12.5 (± 3.64) | 7.88 (± 0.43) | 3.25 (± 0.52) | 5.86 (± 0.44) |
| | EDITS | OOM | OOM | **0.18** (± 1.78) | 10.7 (± 0.66) | 7.36 (± 0.05) | OOM | OOM |
| | FairEdit | OOM | OOM | 3.15 (± 3.73) | **1.95** (± 0.21) | 7.39 (± 0.50) | OOM | OOM |
| | FairVGNN | 6.33 (± 1.90) | 5.31 (± 1.19) | 3.13 (± 0.28) | 9.93 (± 0.88) | 6.54 (± 0.53) | OOM | OOM |

for similar individuals. We then analyze the trade-offs between different fairness criteria (RQ3) and evaluate the computational efficiency of these methods (RQ4). The findings provide valuable insights into the strengths and limitations of each method, guiding the selection of appropriate fairness-aware models for practical use. Due to space limit, we present a subset of the benchmarking results in this section, and the complete results are discussed in Appendix.

### 4.1 PERFORMANCE UNDER GROUP FAIRNESS (RQ1)

We first perform experiments to answer RQ1. Specifically, we present the quantitative results corresponding to those graph learning methods focusing on group fairness in Table 2. Note that we present the results on AUC-ROC score (utility) and $\Delta_{\text{SP}}$ (fairness) as an example, and the complete results are in Appendix. Here DeepWalk and GNN are added as baselines for shallow embedding methods and GNN-based methods, respectively. We observe that different methods yield different levels of trade-offs between utility and

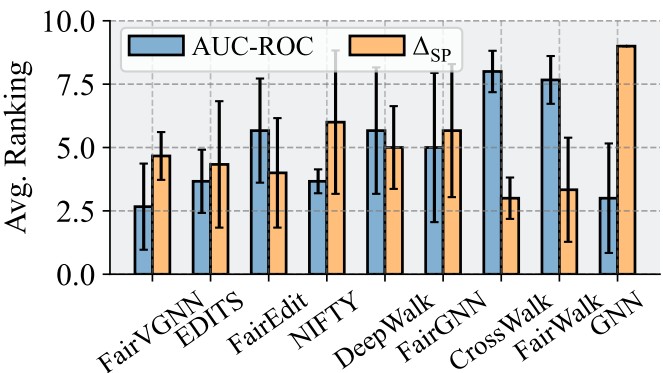

Figure 2: Average rankings on AUC-ROC score and $\Delta_{\text{SP}}$ across all datasets. Methods are ranked in ascending order by the summation of two rankings.

fairness. To better understand the strengths and limitations associated with each algorithm, we calculate the average ranking of each method on datasets free from OOM. We show their average rankings (ordered by the summation of two average rankings) in Figure 2.

**Finding 1: Fairness-aware graph learning methods excel differently on group fairness.** According to Table 2 and Figure 2, we found that different fairness-aware graph learning methods exhibit different types of proficiency between utility and fairness. Specifically, we have the following observations. First, top-ranked methods (those ranked at the left in Figure 2)

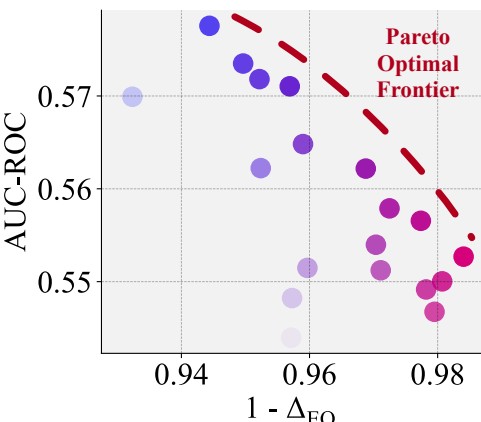

Figure 3: Pareto optimal frontier between AUC-ROC score and $\Delta_{\text{EO}}$ from FairGNN on Credit Default.

are all GNN-based ones. This verifies the natural advantage of GNNs in achieving both accurate and fair predictions owing to their superior fitting ability. Second, fairness-aware shallow embedding methods (i.e., CrossWalk and FairWalk) yield the top-ranked performances in terms of fairness. Considering that these shallow embedding methods do not take node attributes as input compared with those GNN-based ones, such an observation can be partially attributed to the absence of bias encoded in the node attributes. Third, neither DeepWalk nor GNN yields top-ranked performance under utility. This implies that improving fairness does not necessarily jeopardize utility. This may conflict with the common belief that achieving one necessarily means sacrifices the other, while it aligns with the observations of other representative works in this area (Dai & Wang, 2021; Dong et al., 2022). Additionally, to better characterize the trade-off between utility and accuracy, we show an exemplary (estimated) Pareto optimal frontier between AUC-ROC score and $\Delta_{\text{EO}}$ during hyper-parameter search in Figure 3. We observe that such a frontier implicitly prevents a graph learning model from further improving the performance under both evaluation metrics.

### 4.2 Performance Under Individual Fairness (RQ2)

We then answer RQ2 by comparing the performance of graph machine learning methods focusing on individual fairness. Similar to RQ1, we will explore their performance on both utility and fairness. Specifically, we choose the AUC-ROC score as an exemplary metric for utility, and we adopt the three metrics for individual fairness presented in Section 2 to measure the level of individual fairness. Without loss of generality, we adopt a common setting of $k = 10$ for the ranking-based individual fairness metric NDCG@$k$ (Dong et al., 2021b). We present the experimental results in Table 3, and the complete results with supplementary discussion are given in Appendix.

**Finding 2: Fairness-aware graph learning methods exhibit different levels of versatility on individual fairness.** According to Table 3, we have the following observations. First, in terms of utility, we observe that improving individual fairness typically leads to stronger compromise on utility. The vanilla GNN generally achieves the best utility across most datasets. The collected fairness-aware graph learning methods generally sacrifice a certain level of utility in order to improve the level of individual fairness. Second, in terms of fairness, we observe that these methods exhibit different levels of versatility. Specifically, InFoRM, REDRESS, and GUIDE yield the best performance on those individual fairness goals they are equipped with by design, i.e., Lipschitz-based fairness (measured with $B_{\text{Lipschitz}}$), ranking-based fairness (measured by NDCG@$k$), and ratio-based fairness (measured by GDIF), respectively. However, GUIDE also delivers the second best $B_{\text{Lipschitz}}$ and NDCG@$k$ on four out of the seven datasets at the same time, which makes it the most versatile method among the studied three. This implies that compared with the other two methods, GUIDE contributes to a more general improvement in terms of the levels of individual fairness instead of only optimizing one objective and sacrificing others. Such an advantage can be attributed to the compositional design of its objective function, which consists of different fairness objectives (Song et al., 2022). Similar versatility is also observed in InFoRM, which yields the second-best performance on GDIF in three out of the seven datasets. Hence, we conclude that these methods exhibit different levels of versatility under individual fairness.

### 4.3 Trade-off Between Different Fairness Criteria (RQ3)

We now answer RQ3 by comparing the performance of fairness-aware graph learning methods under different fairness metrics. Considering the scarcity of methods under individual fairness, here we focus on group fairness and discuss the results over individual fairness in Appendix. Specifically, for each of the three group fairness metrics given in Section 2, we calculate the average ranking of each method on those datasets free from OOM, and we present the comparison of their average rankings

Table 3: Comparison of graph learning methods focusing on individual fairness. Note that results include AUC-ROC score, $B_{\text{Lipschitz}}$, NDCG@$k$, and GDIF; complete results are in Appendix. The best ones are in **bold**; the second best ones are underlined; OOM denotes out-of-memory.

| Metrics | Models | Pokec-z | Pokec-n | German Credit | Credit Defaulter | Recidivism | AMiner-S | AMiner-L |
|---|---|---|---|---|---|---|---|---|
| **AUC-ROC** | GNN | **66.50** ($\pm$0.53) | **67.62** ($\pm$0.76) | **69.70** ($\pm$3.48) | 62.29 ($\pm$4.85) | **82.47** ($\pm$1.41) | **82.23** ($\pm$0.56) | **88.15** ($\pm$0.11) |
| | InFoRM | 60.53 ($\pm$3.67) | 64.12 ($\pm$4.12) | 63.61 ($\pm$4.93) | 62.72 ($\pm$5.87) | 79.66 ($\pm$6.58) | 69.75 ($\pm$5.18) | 73.72 ($\pm$7.97) |
| | REDRESS | 62.31 ($\pm$6.52) | 64.70 ($\pm$4.88) | 63.79 ($\pm$4.40) | 64.39 ($\pm$5.25) | 69.52 ($\pm$5.58) | OOM | OOM |
| | GUIDE | 63.55 ($\pm$3.62) | 60.36 ($\pm$4.43) | 65.56 ($\pm$4.18) | 64.64 ($\pm$3.86) | 75.09 ($\pm$5.41) | 73.34 ($\pm$4.28) | OOM |
| **$B_{\text{Lipschitz}}$** | GNN | 2.5e6 ($\pm$2e4) | 5.5e3 ($\pm$3e3) | 3.6e3 ($\pm$2e3) | 1.3e4 ($\pm$7e3) | 1.2e7 ($\pm$3e5) | 2.2e6 ($\pm$3e5) | 3.2e7 ($\pm$5e5) |
| | InFoRM | **9.1e2** ($\pm$1e2) | **3.4e3** ($\pm$4e3) | **2.0e2** ($\pm$7e2) | **5.2e1** ($\pm$3e2) | **4.7e3** ($\pm$9e3) | **9.7e3** ($\pm$4e3) | **9.8e4** ($\pm$3e3) |
| | REDRESS | 2.0e5 ($\pm$1e4) | 1.9e5 ($\pm$2e4) | 7.0e3 ($\pm$1e3) | 1.2e4 ($\pm$3e3) | 2.6e4 ($\pm$6e3) | OOM | OOM |
| | GUIDE | 1.8e3 ($\pm$3e2) | 4.0e3 ($\pm$6e2) | 6.4e3 ($\pm$9e2) | 4.2e3 ($\pm$3e2) | 1.1e5 ($\pm$1e4) | 1.5e4 ($\pm$7e3) | OOM |
| **NDCG@$k$** | GNN | 44.56 ($\pm$0.59) | 37.01 ($\pm$0.26) | 31.42 ($\pm$1.49) | 39.01 ($\pm$1.05) | 15.31 ($\pm$0.32) | **43.74** ($\pm$0.70) | **37.75** ($\pm$0.19) |
| | InFoRM | 48.78 ($\pm$3.62) | 44.09 ($\pm$3.00) | 35.89 ($\pm$3.69) | 37.11 ($\pm$3.18) | 19.81 ($\pm$1.74) | 38.85 ($\pm$2.07) | 33.34 ($\pm$1.70) |
| | REDRESS | **54.30** ($\pm$3.08) | **48.53** ($\pm$3.85) | **42.82** ($\pm$3.62) | **42.74** ($\pm$2.11) | **25.30** ($\pm$1.96) | OOM | OOM |
| | GUIDE | 49.02 ($\pm$2.72) | 47.27 ($\pm$4.72) | 32.70 ($\pm$2.02) | 37.38 ($\pm$2.69) | 21.50 ($\pm$2.18) | 39.16 ($\pm$2.26) | OOM |
| **GDIF** | GNN | 111.92 ($\pm$0.81) | 232.16 ($\pm$24.2) | 125.87 ($\pm$11.1) | 166.78 ($\pm$36.1) | 112.78 ($\pm$1.29) | 114.05 ($\pm$1.17) | 112.72 ($\pm$1.21) |
| | InFoRM | 118.07 ($\pm$10.2) | 116.17 ($\pm$5.65) | 136.94 ($\pm$10.3) | 160.62 ($\pm$11.2) | 112.90 ($\pm$8.66) | 125.36 ($\pm$11.4) | 127.84 ($\pm$8.51) |
| | REDRESS | 167.56 ($\pm$7.12) | 124.08 ($\pm$10.8) | 139.98 ($\pm$8.84) | 163.84 ($\pm$5.75) | 109.58 ($\pm$7.33) | OOM | OOM |
| | GUIDE | **108.75** ($\pm$5.89) | **110.58** ($\pm$9.36) | **112.35** ($\pm$8.27) | 149.97 ($\pm$5.14) | **104.17** ($\pm$8.21) | 112.28 ($\pm$7.80) | OOM |

in Figure 4. Generally, a good trade-off indicates that the superiority in one fairness metric does not significantly sacrifice the fairness levels measured by other metrics.

**Finding 3: Fairness-aware graph learning methods struggle for a balance.** According to Figure 4, we have the following observations. First, fairness-aware graph learning methods

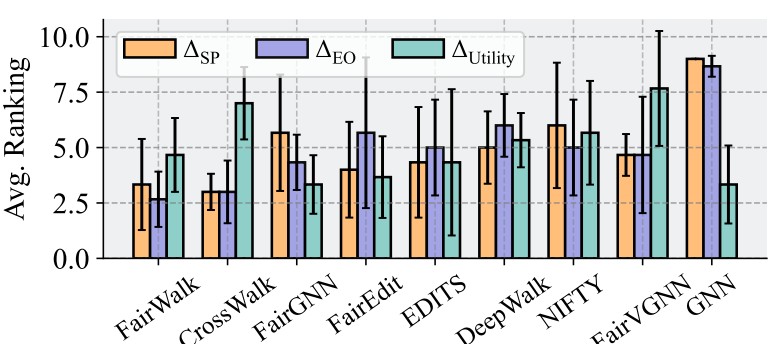

based on shallow embedding methods, i.e., FairWalk and Cross-Walk, generally outperform those GNN-based ones when considering the balance over all three fairness metrics. Notably, they also achieve the best performance on both $\Delta_{\text{SP}}$ and $\Delta_{\text{EO}}$. This aligns with the observations shown in Section 4.1, which can be attributed to the ab-

Figure 4: Average rankings on $\Delta_{\text{SP}}$, $\Delta_{\text{EO}}$, and $\Delta_{\text{Utility}}$ across all datasets. Methods are ranked in ascending order by the summation of average rankings on all three fairness metrics.

sence of bias brought by node attributes. Second, we note that the utility difference-based fairness (measured with $\Delta_{\text{Utility}}$) is not an explicit optimization goal for any of these methods. Despite this, top-ranked fairness-aware methods based on GNNs (e.g., FairGNN and FairEdit) clearly outperform those based on shallow embedding methods in terms of $\Delta_{\text{Utility}}$. This can be attributed to the superior fitting ability of GNNs and informative node attributes, which implicitly helps ensure that no subgroup bears significantly worse performance than others. Based on the above observations, we conclude that these methods always struggle for a balance between different fairness metrics, and one method can hardly do well on all of them.

### 4.4 COMPUTATIONAL EFFICIENCY (RQ4)

Finally, we answer RQ4 by comparing the computational efficiency of the collected fairness-aware graph learning methods. Here, we utilize running time in seconds to measure efficiency, and we also collect the associated utility (measured with AUC-ROC score). We present an exemplary comparison

across all collected graph learning models (two baselines and ten fairness-aware ones) on the Credit Default dataset in Figure 5. The comparison on other datasets is presented and discussed in Appendix.

**Finding 4: Fairness-aware graph learning methods generally sacrifice efficiency.** According to Figure 5, we have the following observations. First, fairness-aware graph learning methods based on GNNs exhibit a clear sacrifice on efficiency, where EDITS under group fairness and REDRESS under individual fairness sacrifice the most. This can be attributed to their computationally expensive optimization strategy: EDITS requires optimizing the whole graph topology, while REDRESS calculates different similarity rankings (across all nodes) in every learning epoch. In contrast to the clear sacrifice on efficiency, we also observe that most fairness-aware graph learning methods maintain a relatively high level of utility,

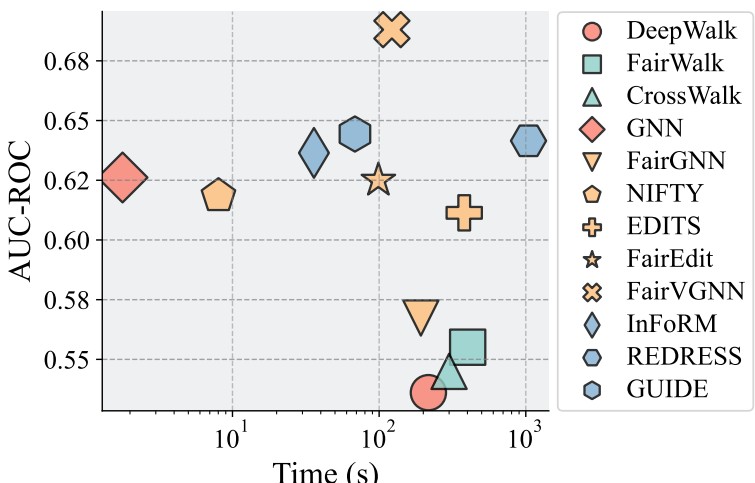

Figure 5: An exemplary comparison of AUC-ROC and running time across different collected graph learning methods on Credit Default dataset.

which remains consistent with the general utility assessment shown in Section 4.1. Second, although those based on shallow embedding methods bear longer running time (than most GNN-based ones), they only marginally sacrifice efficiency. A primary reason is that compared with GNN-based ones, they usually do not introduce much additional computation in the calculation and optimization of the objective function. In fact, both FairWalk and CrossWalk facilitate their fairness levels by adopting different transition probability distributions to perform random walks on graphs. Meanwhile, we also notice that those based on shallow embedding methods generally bear worse utility than the GNN-based ones, which is also consistent with the discussion in Section 4.1. Based on the observations above, we conclude that these fairness-aware methods generally sacrifice efficiency compared with the vanilla baseline methods.

## 5 A GUIDE FOR PRACTITIONERS

Based on the discussion above, we conclude that each fairness-aware graph learning method bears its strengths and limitations from different perspectives. Therefore, it becomes crucial to select the most suitable methods to use carefully. To assist practitioners in making informed decisions in real-world applications, this section provides a guide to help choose the most appropriate fairness-aware graph learning methods such that their strengths can be fully leveraged to address fairness-related concerns while maintaining a proper level of performance.

Specifically, we propose to organize this guide from two perspectives, including group fairness and individual fairness. From the perspective of group fairness, if the main priority is to achieve the best performance on typical group fairness metrics such as $\Delta_{SP}$ and $\Delta_{EO}$, while utility and efficiency are less of a concern, fairness-aware shallow embedding methods including FairWalk and CrossWalk are recommended choices. The reason is that these methods can generally achieve top-ranked performance in terms of group fairness, although the corresponding utility and efficiency are usually inferior to GNN-based methods. If the main priority is to achieve a good balance between utility and group fairness, GNN-based methods such as FairVGNN, EDITS, FairEdit, and NIFTY are recommended. This is because these methods usually achieve a more satisfactory trade-off between utility and group fairness compared with those based on shallow embedding methods. Furthermore, we note that FairGNN maintains a better trade-off among all three fairness metrics, which makes it more suitable for applications with significant emphasis on optimizing different types of fairness. From the perspective of individual fairness, since each method bears a different fairness optimization

goal, we recommend selecting the one with the most desired goal of individual fairness. Meanwhile, we notice that GUIDE achieves a superior balance between $B_{\text{Lipschitz}}$ and GDIF compared with the other two methods. Hence GUIDE is recommended if higher levels of individual fairness is desired.

## 6 RELATED WORKS

**Benchmarking Graph Learning Methods.** Existing studies have explored two mainstream benchmarks for graph learning methods, i.e., usability-oriented ones and trustworthiness-oriented ones. Specifically, usability-oriented ones focus on evaluating models' capabilities in accomplishing specific graph learning tasks, including node classification (Shchur et al., 2018; Izadi et al., 2020; Luan et al., 2021), link prediction (Bordes et al., 2013; Shang et al., 2018; Suchanek et al., 2007), and representation learning (Stier & Granitzer; Ren et al., 2020). In addition to those focusing on utility (e.g., F1-score in node classification tasks), a few existing studies also explored efficiency, such as comparisons on training time (Said et al., 2023) and memory usage (Huang et al., 2023). On the other hand, trustworthiness-oriented ones mainly aim to provide comprehensive analysis on how well graph learning models can be trusted, such as studies from the perspective of robustness (Bojchevski & Günnemann, 2019; Zügner & Günnemann, 2019) and interpretability (Agarwal et al., 2018; Xuanyuan et al., 2023). However, from the perspective of algorithmic fairness, existing benchmarks remain scarce. To the best of our knowledge, Qian et al. (Qian et al., 2024) took an initial step towards developing a fairness-aware graph learning benchmark. However, only two representative works are evaluated in their benchmark, which limits the insights it reveals. Different from the existing research work above, our work serves as an initial step towards a comprehensive benchmark on fairness-aware graph learning methods, which reveals key insights on their strengths and limitations and exhibits the potential to facilitate broader applications.

**Fairness-Aware Graph Learning.** In graph learning tasks, unfairness can be defined with different criteria and exhibited from different perspectives (Dong et al., 2023b). In general, two fairness notions are the most widely discussed ones by existing studies, i.e., group fairness and individual fairness. Specifically, group fairness emphasizes that the learning methods should not yield discriminatory predictions or decisions targeting individuals belonging to any particular sensitive subgroup (race, gender, etc.) (Dwork et al., 2011). Common approaches to mitigate the bias revealed by the notion of group fairness include rebalancing (Khajehnejad et al., 2022; Farnadi et al., 2018; Current et al., 2022; Buyl & Bie, 2021), adversarial learning (Dai & Wang, 2021; Khajehnejad et al., 2020; Xu et al., 2021; Bose & Hamilton, 2019), edge rewiring (Dong et al., 2022; Li et al., 2021; Kose & Shen, 2022; Jalali et al.), and orthogonal projection (Palowitch & Perozzi, 2020; Zeng et al., 2021). On the other hand, *individual fairness* notion requires models to treat similar individuals similarly (Dwork et al., 2011). Existing works that mitigate the bias revealed by individual fairness include optimization with constraints (Gupta & Dukkipati, 2021) and regularizations (Fan et al., 2021; Dong et al., 2021a; Kang et al., 2020a; Lahoti et al., 2019). Other fairness issues have also been studied in recent years Liu et al. (2023); Arun et al. (2023). However, since they have not been widely adopted in real-world applications, they are not the main focus of this paper. Despite the abundant efforts, there still lacks a comprehensive benchmark to facilitate the understanding of those representative fairness-aware graph learning methods. We present a comprehensive benchmark to provide guidance based on the results over a wide range of representative fairness-aware graph learning methods.

## 7 CONCLUSION

In this paper, we introduced a comprehensive benchmark for fairness-aware graph learning methods, which bridges a critical gap between the current literature and broader applications. Specifically, we designed a systematic evaluation protocol, collected ten representative methods, and conducted extensive experiments on seven real-world attributed graph datasets from various domains. Our in-depth analysis revealed key insights into the strengths and limitations of existing methods in terms of group fairness, individual fairness, balancing different fairness criteria, and computational efficiency. These findings, along with the practical guide we provided, offer valuable guidance for practitioners to select appropriate methods based on their specific requirements. While we focused on the node classification task in this paper, evaluations on other graph learning tasks remain a future direction to be explored, which will further enrich the understanding of the performance of these methods and expand their applicability across a wider range of applications.

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

## A    DOCUMENTATION OF NEW DATASETS

**Introduction of New Datasets.** In this benchmark, we introduce two new crafted datasets: AMiner-S and AMiner-L, which are coauthor networks constructed from the AMiner network (Wan et al., 2019) in two different ways. The AMiner-S dataset is extracted from AMiner by its largest connected component, and contains 39,424 nodes, 52,460 edges, and 5,694 attributes in total. Here the nodes denote the researchers, the edges represent the co-authorship between researchers, and the attributes are created from the abstracts of the associated papers. In addition, the sensitive attribute is the continent of the affiliation of each researcher belongs to, and the task associated with this dataset is to predict the primary research field of each researcher. The AMiner-L dataset is extracted from AMiner by random walk, which has 129,726 nodes, 591,039 edges, and 5,694 attributes in total. All the settings including the sensitive attribute and the associated tasks are the same with AMiner-S. It is worth-noting that both datasets AMiner-S and AMiner-L are anonymous and thus have no privacy concern, which ensures compliance with privacy regulations such as the General Data Protection Regulation (GDPR) and allows for broader sharing and usage across institutions.

**Intended Uses.** The two datasets are designed for research in the field of graph learning, especially designed for fairness-related research on graphs. The datasets allow researchers to evaluate their fairness-aware graph learning algorithms, thus empower them to measure and mitigate bias that may exhibit by graph learning algorithms.

## B    REPRODUCIBILITY

In this section, we will introduce the details of the experiments for the purpose of reproducibility. We first provide a detailed description of benchmark datasets employed in this study. Subsequently, we describe the implementation of the experiments on these datasets, followed by an in-depth explanation of code basis and hardware support of the implementation.

**Benchmark Datasets.** We collected seven real-world attributed graph datasets in this benchmark paper, including five existing commonly used ones and two newly constructed ones. We provide a brief introduction for each as follows. *(1) Pokec-z (Takac & Zabovsky, 2012)*. The Pokec-z dataset is sampled from Pokec, which is the most popular on-line social network in Slovakia. Pokec contains anonymized data of the whole social network in 2012, in which the profiles contain gender, age, hobbies, interest, education, working field, etc. Here the region corresponding to each user is considered as the sensitive attributes, and the task is to predict the working field of each user. *(2) Pokec-n (Takac & Zabovsky, 2012)*. The Pokec-n dataset is sampled from Pokec as well, while the users in Pokec-n come from different geographical regions compared with those in Pokec-z. Pokec-n shares the same settings on sensitive attributes and predictive task as those of Pokec-z. *(3) German Credit (Markelle Kelly)*. The German Credit dataset is a credit graph, where nodes represent clients in a German bank and they are connected based on the similarity of their credit accounts. Here the task is to classify the credit risks of clients into high/low, and gender is considered as the sensitive attribute. *(4) Credit Defaulter (Yeh & Lien, 2009)*. Credit contains clients who are connected based on the similarity of their spending and payment patterns. Here the task is to classify whether each client will default on the credit card payment or not, and age is considered as the sensitive attribute. *(5) Recidivism (Jordan & Freiburger, 2015)*. Recidivism dataset is a graph of defendants who got released on bail at the U.S state courts during 1990-2009. These defendants are connected based on the similarity of past criminal records and demographics. The task is to determine whether a defendant deserves bail or not, and their race is considered as the sensitive attribute. *(6) AMiner-S (newly constructed)*. AMiner-S is a co-author graph we extracted from the AMiner network (Wan et al., 2019) by its largest connected component. Here nodes represent the researchers in different fields, and edges denote the co-authorship between researchers. The sensitive attribute is the continent of the affiliation each researcher belongs to, and the task is to predict the primary research field of each researcher. *(5) AMiner-L (newly constructed)*. AMiner-L is a co-author graph we extracted from the AMiner network by random walk. Compared with AMiner-S, AMiner-L bears a larger scale. AMiner-L shares the same settings on sensitive attributes and predictive task as those of AMiner-S.

**Experimental Settings.** All experiments are repeated for three times. For a fair comparison, we perform a grid search to tune hyperparameters for all algorithms. For most of the experiments, we adopt Adam optimizer. All major experiments can be executed with the provided code.

Table 1: Performance comparison between different group fairness-aware graph learning models. The best ones are in **Bold**, and OOM represents out-of-memory.

| Metrics | Models | Pokec-z | Pokec-n | German Credit | Credit Defaulter | Recidivism | AMiner-S | AMiner-L |
|---|---|---|---|---|---|---|---|---|
| **ACC** | DeepWalk | 66.44 (± 1.33) | 63.58 (± 1.06) | 62.52 (± 3.59) | 71.64 (± 0.48) | 90.14 (± 1.52) | 86.35 (± 0.27) | 88.54 (± 1.56) |
| | FairWalk | 64.62 (± 0.34) | 61.08 (± 0.33) | 68.70 (± 0.00) | 69.48 (± 0.27) | 75.32 (± 0.10) | 79.75 (± 0.08) | 96.18 (± 0.01) |
| | CrossWalk | 57.98 (± 0.23) | 61.58 (± 0.27) | 63.85 (± 1.59) | 67.07 (± 0.12) | 88.05 (± 0.10) | 79.73 (± 0.12) | 96.21 (± 0.03) |
| | GNN | 66.04 (± 0.43) | 68.12 (± 1.09) | 66.94 (± 1.01) | 76.41 (± 1.88) | 86.68 (± 2.50) | 89.97 (± 0.62) | 91.21 (± 0.01) |
| | FairGNN | 66.41 (± 1.00) | 66.80 (± 0.52) | 67.32 (± 2.52) | 79.91 (± 14.1) | 90.77 (± 2.98) | 92.22 (± 0.18) | OOM |
| | NIFTY | 63.35 (± 0.13) | 68.26 (± 0.81) | 64.13 (± 2.16) | 65.55 (± 0.09) | 86.49 (± 0.74) | 89.27 (± 0.03) | 91.92 (± 0.19) |
| | EDITS | OOM | OOM | 62.25 (± 4.95) | 71.02 (± 0.34) | 90.99 (± 0.14) | OOM | OOM |
| | FairEdit | OOM | OOM | 65.93 (± 2.84) | 71.01 (± 3.63) | 83.86 (± 0.44) | OOM | OOM |
| | FairVGNN | 67.52 (± 3.77) | 68.09 (± 0.51) | 70.10 (± 0.58) | 78.66 (± 4.29) | 85.69 (± 5.37) | OOM | OOM |
| **AUC-ROC Score** | DeepWalk | 66.50 (± 1.34) | 61.85 (± 1.06) | 56.90 (± 1.75) | 53.61 (± 0.66) | 87.18 (± 1.34) | 73.58 (± 0.43) | 82.68 (± 3.28) |
| | FairWalk | 64.92 (± 0.43) | 61.52 (± 0.34) | 54.05 (± 0.83) | 55.51 (± 0.29) | 72.09 (± 0.11) | 65.35 (± 0.54) | 88.72 (± 0.08) |
| | CrossWalk | 58.99 (± 0.27) | 62.98 (± 0.27) | 51.42 (± 0.43) | 54.50 (± 0.42) | 82.89 (± 0.11) | 64.44 (± 0.52) | 89.67 (± 0.04) |
| | GNN | 64.16 (± 0.62) | 67.05 (± 1.14) | 67.36 (± 3.59) | 62.62 (± 0.51) | 84.60 (± 2.10) | 81.95 (± 1.46) | 86.82 (± 0.11) |
| | FairGNN | 69.47 (± 1.04) | 68.51 (± 0.51) | 52.91 (± 2.15) | 56.73 (± 3.16) | 92.87 (± 2.42) | 86.23 (± 0.14) | OOM |
| | NIFTY | 62.58 (± 0.14) | 66.78 (± 0.82) | 62.94 (± 5.78) | 61.85 (± 0.70) | 85.58 (± 0.83) | 79.28 (± 0.15) | 86.62 (± 0.69) |
| | EDITS | OOM | OOM | 60.02 (± 1.10) | 61.14 (± 0.36) | 92.34 (± 0.31) | OOM | OOM |
| | FairEdit | OOM | OOM | 56.30 (± 2.33) | 62.50 (± 0.61) | 81.97 (± 0.48) | OOM | OOM |
| | FairVGNN | 71.19 (± 0.94) | 70.14 (± 0.55) | 65.48 (± 3.46) | 68.81 (± 0.81) | 84.74 (± 2.70) | OOM | OOM |
| $\Delta_{\text{SP}}$ | DeepWalk | 5.49 (± 1.07) | 5.90 (± 0.88) | 10.4 (± 1.01) | 6.69 (± 0.31) | 6.50 (± 0.18) | 6.75 (± 0.29) | 6.41 (± 0.46) |
| | FairWalk | 0.60 (± 1.89) | 0.29 (± 2.12) | 3.36 (± 1.01) | 6.20 (± 0.32) | 4.67 (± 0.33) | 3.06 (± 0.32) | 4.28 (± 0.17) |
| | CrossWalk | 1.75 (± 1.17) | 0.21 (± 1.63) | 0.35 (± 1.75) | 6.35 (± 0.51) | 5.14 (± 0.21) | 3.59 (± 0.43) | 5.60 (± 0.42) |
| | GNN | 10.4 (± 1.46) | 14.7 (± 0.40) | 32.4 (± 1.93) | 20.6 (± 4.34) | 8.54 (± 0.10) | 7.28 (± 0.31) | 6.75 (± 0.00) |
| | FairGNN | 2.06 (± 1.82) | 8.11 (± 1.16) | 14.2 (± 0.83) | 2.51 (± 5.61) | 7.48 (± 0.30) | 5.36 (± 0.27) | OOM |
| | NIFTY | 2.48 (± 0.47) | 2.42 (± 0.84) | 0.26 (± 0.41) | 12.5 (± 3.64) | 7.88 (± 0.43) | 3.25 (± 0.52) | 5.86 (± 0.44) |
| | EDITS | OOM | OOM | 0.18 (± 1.78) | 10.7 (± 0.66) | 7.36 (± 0.05) | OOM | OOM |
| | FairEdit | OOM | OOM | 3.15 (± 3.73) | 1.95 (± 0.21) | 7.39 (± 0.50) | OOM | OOM |
| | FairVGNN | 6.33 (± 1.90) | 5.31 (± 1.19) | 3.13 (± 0.28) | 9.93 (± 0.88) | 6.54 (± 0.53) | OOM | OOM |
| $\Delta_{\text{EO}}$ | DeepWalk | 7.31 (± 1.19) | 4.85 (± 2.07) | 13.7 (± 2.17) | 5.79 (± 1.21) | 4.48 (± 0.38) | 11.5 (± 1.47) | 11.1 (± 3.11) |
| | FairWalk | 0.20 (± 1.35) | 0.08 (± 2.47) | 2.68 (± 0.86) | 4.40 (± 0.64) | 1.34 (± 1.03) | 2.54 (± 1.90) | 2.44 (± 1.01) |
| | CrossWalk | 1.27 (± 0.96) | 1.46 (± 1.10) | 4.59 (± 1.83) | 1.16 (± 0.60) | 1.70 (± 0.50) | 2.23 (± 0.79) | 4.50 (± 1.77) |
| | GNN | 8.99 (± 1.07) | 17.2 (± 1.13) | 23.4 (± 1.48) | 19.2 (± 4.41) | 6.85 (± 0.23) | 12.3 (± 0.65) | 8.87 (± 0.22) |
| | FairGNN | 0.29 (± 1.06) | 9.84 (± 0.98) | 9.31 (± 0.03) | 1.62 (± 5.94) | 3.60 (± 0.24) | 6.26 (± 0.60) | 6.63 (± 0.22) |
| | NIFTY | 3.25 (± 0.47) | 6.17 (± 0.88) | 3.37 (± 0.45) | 9.89 (± 3.73) | 3.14 (± 0.24) | 0.70 (± 1.54) | 6.63 (± 0.22) |
| | EDITS | OOM | OOM | 2.19 (± 7.06) | 7.74 (± 0.48) | 4.63 (± 0.53) | OOM | OOM |
| | FairEdit | OOM | OOM | 10.1 (± 2.95) | 0.94 (± 0.24) | 7.04 (± 0.63) | OOM | OOM |
| | FairVGNN | 2.41 (± 2.09) | 7.61 (± 0.85) | 1.80 (± 0.10) | 7.34 (± 0.39) | 5.62 (± 0.45) | OOM | OOM |
| $\Delta_{\text{Utility}}$ | DeepWalk | 3.38 (± 1.48) | 0.21 (± 1.27) | 17.0 (± 4.27) | 6.61 (± 0.96) | 0.68 (± 1.53) | 0.95 (± 0.39) | 2.28 (± 1.14) |
| | FairWalk | 4.04 (± 0.36) | 0.24 (± 0.40) | 10.6 (± 1.05) | 2.33 (± 0.39) | 1.95 (± 0.32) | 0.39 (± 0.32) | 1.43 (± 0.10) |
| | CrossWalk | 0.93 (± 0.26) | 2.86 (± 0.46) | 16.8 (± 2.18) | 9.27 (± 0.46) | 1.96 (± 0.16) | 5.10 (± 0.20) | 1.88 (± 0.12) |
| | GNN | 2.19 (± 1.09) | 2.57 (± 1.21) | 2.90 (± 1.19) | 4.63 (± 2.63) | 1.26 (± 2.66) | 0.33 (± 0.74) | 1.60 (± 0.04) |
| | FairGNN | 0.18 (± 1.02) | 0.02 (± 0.60) | 9.19 (± 2.71) | 1.42 (± 12.13) | 1.52 (± 2.96) | 3.72 (± 0.15) | OOM |
| | NIFTY | 0.64 (± 0.30) | 1.85 (± 0.82) | 3.35 (± 3.25) | 3.75 (± 0.98) | 4.60 (± 0.77) | 4.70 (± 0.09) | 4.22 (± 0.16) |
| | EDITS | OOM | OOM | 3.06 (± 5.28) | 11.6 (± 2.25) | 0.29 (± 0.17) | OOM | OOM |
| | FairEdit | OOM | OOM | 0.44 (± 4.66) | 2.10 (± 4.61) | 2.75 (± 0.43) | OOM | OOM |
| | FairVGNN | 6.79 (± 3.65) | 0.87 (± 0.64) | 17.1 (± 0.68) | 6.71 (± 3.63) | 2.31 (± 6.06) | OOM | OOM |

**Graph Learning Models.** We implement all fairness-aware graph learning algorithms with their official open-source code. The graph learning models and their associated code links are listed below.

- *FairWalk*: https://github.com/EnderGed/Fairwalk.
- *CrossWalk*: https://github.com/ahmadkhajehnejad/CrossWalk.
- *FairGNN*: https://github.com/EnyanDai/FairGNN.
- *NIFTY*: https://github.com/chirag126/nifty. Under MIT license.
- *EDITS*: https://github.com/yushundong/EDITS.
- *FairEdit*: https://github.com/royull/FairEdit.
- *FairVGNN*: https://github.com/yuwvandy/fairvgnn.
- *InFoRM*: https://github.com/jiank2/inform. Under MIT license.
- *REDRESS*: https://github.com/yushundong/REDRESS.
- *GUIDE*: https://github.com/mikesong724/GUIDE. Under MIT license.

**Hardware.** We conduct all experiments with NVIDIA A6000 GPU (48GB memory), AMD EPYC CPU (2.87 GHz), and 512GB of RAM.

Table 2: Performance comparison between different individual fairness-aware graph learning models. The best ones are in **Bold**, and OOM represents out-of-memory.

| Metrics | Models | Pokec-z | Pokec-n | German Credit | Credit Defaulter | Recidivism | AMiner-S | AMiner-L |
|---|---|---|---|---|---|---|---|---|
| ACC | GNN | **65.71** (± 0.54) | **68.50** (± 0.77) | **67.13** (± 2.12) | **74.43** (± 3.60) | **84.55** (± 1.56) | **90.06** (± 0.32) | **93.12** (± 0.11) |
| | InFoRM | 61.87 (± 3.75) | 66.08 (± 3.66) | 59.40 (± 5.09) | 69.36 (± 5.21) | 80.52 (± 5.94) | 87.41 (± 5.61) | 89.05 (± 5.21) |
| | REDRESS | 62.35 (± 4.28) | 65.39 (± 4.62) | 63.08 (± 4.86) | 68.12 (± 4.70) | 77.77 (± 7.35) | OOM | OOM |
| | GUIDE | 62.86 (± 5.54) | 63.01 (± 5.63) | 60.93 (± 3.91) | 67.07 (± 6.67) | 80.06 (± 5.20) | 87.66 (± 5.46) | OOM |
| AUC-ROC Score | GNN | **66.50** (± 0.53) | **67.62** (± 0.76) | **69.70** (± 3.48) | 62.29 (± 4.85) | **82.47** (± 1.41) | **82.23** (± 0.56) | **88.15** (± 0.11) |
| | InFoRM | 60.53 (± 3.67) | 64.12 (± 4.12) | 63.61 (± 4.93) | 62.72 (± 5.87) | 79.66 (± 6.58) | 69.75 (± 5.18) | 73.72 (± 7.97) |
| | REDRESS | 62.31 (± 6.52) | 64.70 (± 4.88) | 63.79 (± 4.40) | 64.39 (± 5.25) | 69.52 (± 5.58) | OOM | OOM |
| | GUIDE | 63.55 (± 3.62) | 60.36 (± 4.43) | 65.56 (± 4.18) | **64.64** (± 3.86) | 75.09 (± 5.41) | 73.34 (± 4.28) | OOM |
| $\Delta_{SP}$ | GNN | 10.19 (± 0.26) | 14.27 (± 0.36) | **32.97** (± 5.82) | 20.51 (± 1.37) | 8.56 (± 0.37) | 7.24 (± 0.11) | 6.57 (± 0.13) |
| | InFoRM | **4.74** (± 0.50) | **6.62** (± 0.42) | 34.15 (± 3.67) | 25.97 (± 0.79) | 7.57 (± 0.26) | **1.45** (± 0.24) | **2.12** (± 0.39) |
| | REDRESS | 16.35 (± 0.65) | 23.75 (± 1.58) | 42.28 (± 1.62) | **10.84** (± 0.65) | **2.01** (± 0.58) | OOM | OOM |
| | GUIDE | 15.81 (± 0.22) | 13.23 (± 0.67) | 39.18 (± 1.78) | 17.34 (± 1.07) | 4.58 (± 0.44) | 2.15 (± 0.41) | OOM |
| $\Delta_{EO}$ | GNN | 9.20 (± 0.87) | 17.47 (± 0.47) | **23.34** (± 7.89) | 18.76 (± 1.29) | 6.98 (± 1.20) | 12.07 (± 1.77) | 8.98 (± 0.47) |
| | InFoRM | **3.02** (± 0.24) | **9.13** (± 0.43) | 30.22 (± 3.38) | 26.21 (± 1.11) | 5.45 (± 0.33) | **2.58** (± 0.21) | **5.69** (± 0.26) |
| | REDRESS | 20.64 (± 0.28) | 26.01 (± 1.42) | 37.57 (± 1.99) | **10.38** (± 0.32) | **0.10** (± 0.62) | OOM | OOM |
| | GUIDE | 10.93 (± 0.30) | 20.45 (± 0.90) | 36.74 (± 1.43) | 16.72 (± 1.50) | 2.29 (± 0.23) | 3.26 (± 0.22) | OOM |
| $\Delta_{Utility}$ | GNN | 2.37 (± 0.63) | 2.68 (± 0.70) | **1.44** (± 3.79) | 4.84 (± 3.52) | **0.12** (± 1.58) | **0.99** (± 0.33) | 1.03 (± 0.16) |
| | InFoRM | 2.45 (± 4.76) | 4.86 (± 5.50) | 7.63 (± 5.54) | 13.87 (± 4.04) | 0.46 (± 5.82) | 8.42 (± 7.59) | **0.46** (± 7.31) |
| | REDRESS | 8.65 (± 5.87) | **1.85** (± 5.17) | 8.14 (± 5.70) | 7.79 (± 4.81) | 4.60 (± 7.09) | OOM | OOM |
| | GUIDE | **1.73** (± 4.84) | 5.55 (± 4.23) | 12.63 (± 3.23) | 10.99 (± 6.08) | 1.84 (± 5.59) | 1.36 (± 5.53) | OOM |
| $B_{Lipschitz}$ | GNN | 2.5e6 (± 2e4) | 5.5e3 (± 3e3) | 3.6e3 (± 2e3) | 1.3e4 (± 7e3) | 1.2e7 (± 3e5) | 2.2e6 (± 3e5) | 3.2e7 (± 5e5) |
| | InFoRM | **9.1e2** (± 1e2) | **3.4e3** (± 4e3) | **2.0e2** (± 7e2) | **5.2e1** (± 3e2) | **4.7e3** (± 9e3) | **9.7e3** (± 4e3) | **9.8e4** (± 3e3) |
| | REDRESS | 2.0e5 (± 1e4) | 1.9e5 (± 2e4) | 7.0e3 (± 1e3) | 1.2e4 (± 3e4) | 2.6e4 (± 6e3) | OOM | OOM |
| | GUIDE | 1.8e3 (± 3e2) | 4.0e3 (± 6e2) | 6.4e3 (± 9e2) | 4.2e3 (± 3e2) | 1.1e5 (± 1e4) | 1.5e4 (± 7e3) | OOM |
| NDCG@$k$ | GNN | 44.56 (± 0.59) | 37.01 (± 0.26) | 31.42 (± 1.49) | 39.01 (± 1.05) | 15.31 (± 0.32) | **43.74** (± 0.70) | **37.75** (± 0.19) |
| | InFoRM | 48.78 (± 3.62) | 44.09 (± 3.00) | 35.89 (± 3.69) | 37.11 (± 3.18) | 19.81 (± 1.74) | 38.85 (± 2.07) | 33.34 (± 1.70) |
| | REDRESS | **54.30** (± 3.08) | **48.53** (± 3.85) | **42.82** (± 3.62) | **42.74** (± 2.11) | **25.30** (± 1.96) | OOM | OOM |
| | GUIDE | 49.02 (± 2.72) | 47.27 (± 4.72) | 32.70 (± 2.02) | 37.38 (± 2.69) | 21.50 (± 2.18) | 39.16 (± 2.26) | OOM |
| GDIF | GNN | 111.92 (± 0.81) | 232.16 (± 24.2) | 125.87 (± 11.1) | 166.78 (± 36.1) | 112.78 (± 1.29) | 114.05 (± 1.17) | 112.72 (± 1.21) |
| | InFoRM | 118.07 (± 10.2) | 116.17 (± 5.65) | 136.94 (± 10.3) | 160.62 (± 11.2) | 112.90 (± 8.66) | 125.36 (± 11.4) | 127.84 (± 8.51) |
| | REDRESS | 167.56 (± 7.12) | 124.08 (± 10.8) | 139.98 (± 8.84) | 163.84 (± 5.75) | 109.58 (± 7.33) | OOM | OOM |
| | GUIDE | **108.75** (± 5.89) | **110.58** (± 9.36) | **112.35** (± 8.27) | 149.97 (± 5.14) | **104.17** (± 8.21) | **112.28** (± 7.80) | OOM |

**Dependencies.** We list all major packages and their associated versions in our implementation.

- python == 3.9.0
- pygdebias == 1.1.1
- torch == 1.12.0+cu116
- torch-cluster == 1.6.0
- torch-geometric == 2.1.0
- cuda == 11.6
- pandas == 1.4.3
- numpy == 1.22.4
- networkx == 2.8.5
- dgl == 1.1.2+cu116
- scikit-learn == 1.1.1
- scipy == 1.9.0

## C  SUPPLEMENTARY RESULTS & DISCUSSION

**Supplementary Discussion on RQ1.** We perform additional experiments on graph learning algorithms focusing on group fairness and present the results in Table 1 and Figure 1. According to the comprehensive empirical results, we have the consistent observation that different fairness-aware graph learning methods show different advantages in balancing utility and fairness. Specifically, we observe that GNN-based algorithms are among the highest-ranking methods (e.g., most top-ranked results come from GNN-based methods), which verifies the advantage of GNNs in achieving both utility and fairness objectives due to their exceptional fitting ability. In addition, fairness-aware shallow embedding methods achieve the best performance with respect to fairness, especially on the

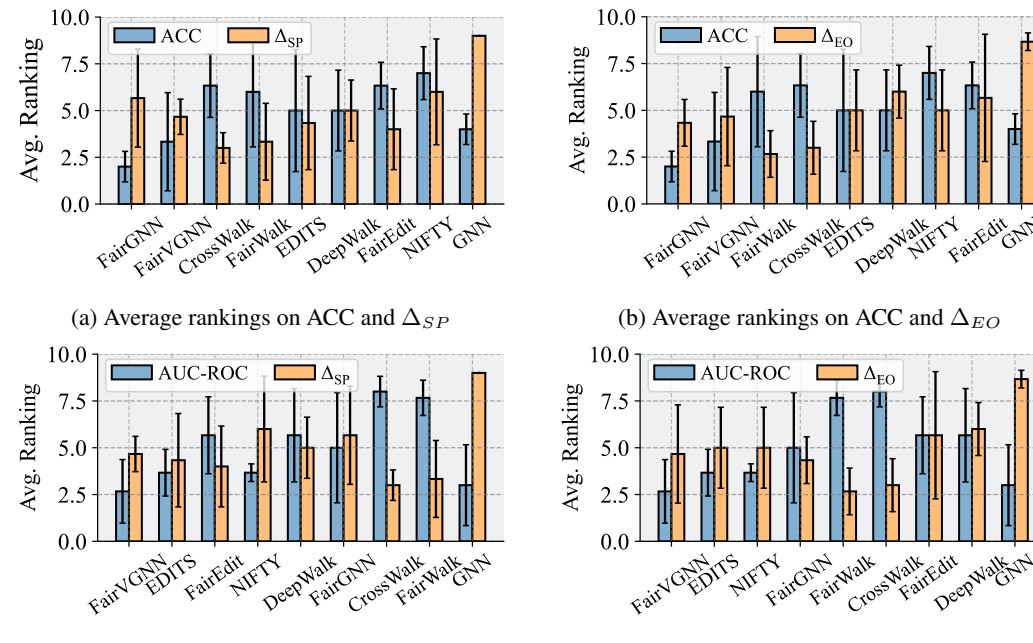

(a) Average rankings on ACC and $\Delta_{SP}$

(b) Average rankings on ACC and $\Delta_{EO}$

(c) Average rankings on AUC-ROC and $\Delta_{SP}$

(d) Average rankings on AUC-ROC and $\Delta_{EO}$

Figure 1: Average rankings on ACC, AUC-ROC, $\Delta_{\text{SP}}$, and $\Delta_{\text{EO}}$ across all datasets. Methods are ranked in ascending order by the summation of average rankings on all three fairness metrics.

traditional fairness metrics $\Delta_{SP}$ and $\Delta_{EO}$. This observation is mainly attributed to the absence of bias encoded in the node attributes. All observations above align with the observations reported in Section 4.1, which demonstrate the consistency over the performance of the studied fairness-aware graph learning methods on a wide range of datasets and metrics.

**Supplementary Discussion on RQ2.** We provide additional results on the performance of graph learning methods focusing on individual fairness in Table 2. We have the following observations. First, all fairness-aware graph learning methods generally sacrifice a certain degree of utility so as to improve the level of individual fairness. Second, different methods exhibit different levels of versatility, where GUIDE can yield competitive performance on more combinations of datasets and metrics compared to other methods. All these observations above align with the finding in the main text, which demonstrate the consistency over a wide range of datasets and metrics.

**Supplementary Discussion on RQ3.** We further discuss RQ3 based on the additional results in Table 1 and Table 2. Specifically, we find that shallow embedding methods and GNN-based methods excel on different fairness metrics, which also implies a struggle for a balance between different fairness metrics. This observation align with the finding in the main text, which demonstrate the consistency over a wide range of datasets and metrics.

**Supplementary Discussion on RQ4.** We present additional results on the computational efficiency of the collected fairness-aware graph learning methods in Figure 2. We found that fairness-aware graph learning methods generally sacrifice efficiency. Specifically, GNN-based fairness-aware graph learning algorithms exhibit a clear sacrifice on efficiency, which can be attributed to their computationally expensive optimization strategy. Meanwhile, algorithms based on shallow embedding methods only marginally sacrifice efficiency, which is resulted from the marginal improvement of additional computation burden in the optimization process compared to those GNN-based ones. All observations above align with the finding in the main text, which demonstrate the consistency over a wide range of datasets.

## D BROADER IMPACTS

This paper presents a comprehensive benchmark for fairness-aware graph learning, which provides extensive evaluation and comparison of existing fairness-aware graph learning algorithms. As a result, we provide insights and guidance to empower better fairness-aware graph learning algorithms in the future, and help facilitate broader applications such as financial lending (Song et al., 2023; Li et al.,

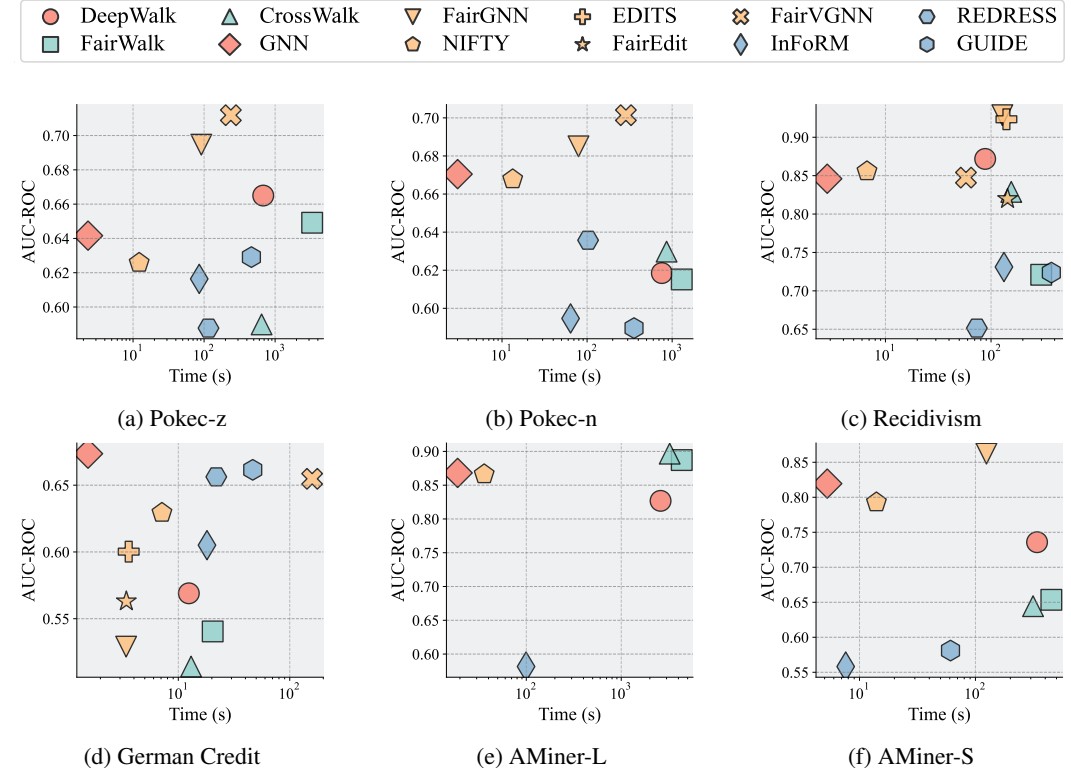

Figure 2: The comparison of AUC-ROC and running time across different fairness-aware graph learning methods on different datasets.

2020), healthcare decision making (Dai et al., 2022; Anderson & Visweswaran, 2024), and policy making (He et al., 2024). At the same time, we note that our work does not have significant negative social impacts we feel necessary to mention here.

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
