# OpenReview forum: "Fairness-Aware Graph Learning: A Benchmark"
_ICLR.cc/2025/Conference — Submitted to ICLR 2025_

### Official Review · Reviewer_fs6n · 2024-11-01

**Soundness:** 3
**Presentation:** 4
**Contribution:** 3
**Rating:** 6
**Confidence:** 5

**Summary:**

This paper presents benchmarking results for fairness corrections to graph learning algorithms. They investigate multiple aspects of 10 baseline methods on node classification tasks.

**Strengths:**

The paper is very well-written and well-organized. The paper tackles a useful problem, benchmarking fairness approaches for graph learning algorithms, which to my knowledge has not been done.

**Weaknesses:**

There are some key weaknesses:

1. The paper only covers node classification, limiting its impact and informativeness.

2. The paper claims to "design a systematic evaluation protocol", but this amounts only to running 10 baselines on standard fairness datasets for graph algorithms. As best I can tell, there is not a novel architecture/system design that the benchmarking experiments depend on.

3. There are some confusions about the results that I point out in my questions.

**Questions:**

Small typo in RQ1: $\Delta_{\text{Utility}}$ should not be called a fairness metric.

RQ3 is written somewhat confusingly -- its not clear what this RQ is investigating. I also don't know what "Utility" is in Figure 4. Can this be elaborated?

Finding 1 (the answer to RQ1) reads "This verifies the natural advantage of GNNs in achieving both accurate and fair predictions owing to their superior fitting ability", suggesting that GNNs best balance group fairness and accuracy. But Finding 3 (the answer to RQ3) suggests DeepWalk-based methods balance all three metrics the best. Combined with the fact that "Utility" from Finding 3 does not seem to be AUC-ROC, this makes these two findings hard to reconcile and compare. How can a practitioner make decisions based on these findings?

L482: This part should be specific that GNNs only exceed at balancing utility & *group* fairness, not overall fairness.

L483: Where do you show that "FairGNN maintains better trade-off between all three fairness metrics"?

S5 should have a disclaimer that these recommendations only hold for node classification.

---

> ### Author Response · Authors · 2024-11-23
> **Authors’ Response 1/2**
>
> We sincerely appreciate the time and efforts you've dedicated to reviewing and providing invaluable feedback to enhance the quality of this paper. We provide a point-to-point reply below for the mentioned concerns and questions.
>
> ---
>
> >  **Reviewer**: The paper only covers node classification, limiting its impact and informativeness.
>
> **Authors**: We thank the reviewer for raising this question. We note that **most traditional fairness metrics**, such as $\Delta_{SP}$ and $\Delta_{EO}$, **do not naturally support tasks other than classification**. Such compatibility has been widely acknowledged by this line of research — as a consequence, we choose the most widely adopted node classification task to conduct our benchmark to ensure broader applicability to both research and applications. We agree with the reviewer that having a benchmark that can be used for various tasks is attractive, while this is a difficult task since **it requires fundamental changes to the commonly used fairness definition and metrics**. This has gone beyond the main goal of this paper, and we will leave it as future work. We hope this helps address your concern.
>
> ---
>
> >  **Reviewer**: The paper claims to "design a systematic evaluation protocol", but this amounts only to running 10 baselines on standard fairness datasets for graph algorithms. As best I can tell, there is not a novel architecture/system design that the benchmarking experiments depend on.
>
> **Authors**: We thank the reviewer for the feedback. We would like to note that we mainly **focus on the most widely used and impactful works** in the area of fairness-aware graph learning, which can directly benefit practitioners and researchers. We will also incorporate the mentioned new works in our future works. We hope this helps address your concern.
>
> Additionally, we would like to highlight that the proposed evaluation strategy is novel: to the best of our knowledge, this is **the first study** that has included a comprehensive set of the **10 most influential fairness-aware graph learning methods in this line of research**. As a comparison, the closest related work evaluates a total number of **two** graph learning methods [1]. To the best of our knowledge, these fairness-aware graph learning methods have not been evaluated under a consistent evaluation protocol under multiple types of fairness metrics and efficiency.
>
> We hope this clarifies the novelty in our evaluation protocol in this area and also helps address your concern.
>
> [1] Qian, X., Guo, Z., Li, J., Mao, H., Li, B., Wang, S., & Ma, Y. (2024, August). Addressing shortcomings in fair graph learning datasets: Towards a new benchmark. In Proceedings of the 30th ACM SIGKDD Conference on Knowledge Discovery and Data Mining (pp. 5602-5612).
>
> ---
>
> >  **Reviewer**: There are some confusions about the results that I point out in my questions.
>
> **Authors**: We thank you for pointing this out. We recognize the need to revise certain expressions to avoid confusion and misunderstanding. We have provided point-to-point responses below, and we will revise our manuscript accordingly to present clearer clarification. We hope this helps address your concern.
>
> ---
>
> >  **Reviewer**: Small typo in RQ1: $\Delta_{Utility}$ should not be called a fairness metric.
>
> **Authors**: We thank you for the feedback. We would like to clarify a misunderstanding here: the metric $\Delta_{Utility}$ measures **the performance gap of a graph learning algorithm between demographic subgroups** (e.g., males vs. females). Therefore, we define $\Delta_{Utility}$ as a metric of fairness, and we introduced the definition of $\Delta_{Utility}$ on page 3 at lines 150 - 153.
>
> We will expand the discussion in Section 2 to further elaborate on $\Delta_{Utility}$ to avoid further misunderstanding. We hope this helps address your concern.
>
> ---
>
> >  **Reviewer**: RQ3 is written somewhat confusingly -- its not clear what this RQ is investigating. I also don't know what "Utility" is in Figure 4. Can this be elaborated?
>
> **Authors**: We thank you for pointing this out. We would like to clarify that "Utility" means the performance on downstream tasks, such as accuracy and AUC-ROC score in node classification tasks. We will expand the discussion in Section 2 and Section 3 to clarify the definition of utility. We hope this helps address your concern.

---

> ### Author Response · Authors · 2024-11-23
> **Authors’ Response 2/2**
>
> ---
>
> >  **Reviewer**: Finding 1 (the answer to RQ1) reads "This verifies the natural advantage of GNNs in achieving both accurate and fair predictions owing to their superior fitting ability", suggesting that GNNs best balance group fairness and accuracy. But Finding 3 (the answer to RQ3) suggests DeepWalk-based methods balance all three metrics the best. Combined with the fact that "Utility" from Finding 3 does not seem to be AUC-ROC, this makes these two findings hard to reconcile and compare. How can a practitioner make decisions based on these findings?
>
> **Authors**: We thank the reviewer for pointing this out, and we acknowledge the need to further clarify the definition of utility and the two findings.
>
> **(1) Definition of utility:** In this work, utility is defined as the performance on downstream tasks, such as accuracy and AUC-ROC score in node classification tasks. We will expand the introduction in Section 2 and 3 to clarify the definition of utility.
>
> **(2) The consistency between the mentioned two findings:**  In finding 1, we found that GNNs enjoy a stronger capability in balancing group fairness and accuracy. However, in finding 2, we observed that DeepWalk-based methods balance all three **Fairness Metrics** better than GNNs. It seems that the misunderstanding comes from the definition of $\Delta_{Utility}$, which is in fact a fairness metric measuring the performance gap of a graph learning algorithm between demographic subgroups. We introduced the definition of $\Delta_{Utility}$ on page 3 at lines 150 - 153, and we will expand the corresponding discussion for clearer clarification.
>
> We hope the detailed explanation above helps address your concern.
>
> ---
>
> >  **Reviewer**: L482: This part should be specific that GNNs only exceed at balancing utility & *group* fairness, not overall fairness.
>
> **Authors**: We thank you for pointing this out. We acknowledge that the reviewer's understanding is correct, and we will expand the discussion to specify the strengths of GNNs on balancing balancing utility & group fairness. We thank you again for your suggestion on improving the quality of our work. We hope this helps address your concern.
>
> ---
>
> >  **Reviewer**: L483: Where do you show that "FairGNN maintains better trade-off between all three fairness metrics"?
>
> **Authors**: We thank the reviewer for raising this point. The observation comes from Figure 4: when all graph learning methods are evaluated based on all three fairness metrics (i.e., $\Delta_{SP}$, $\Delta_{EO}$, $\Delta_{Utility}$), FairGNN has the lowest average ranking across these metrics over all other GNN-based methods. We will ensure the results supporting this claim are clearly referenced in the manuscript and explicitly presented in the corresponding sections. We hope this helps address your concern.
>
> ---
>
> >  **Reviewer**: S5 should have a disclaimer that these recommendations only hold for node classification.
>
> **Authors**: We thank you for pointing this out. Considering that **most traditional fairness metrics**, such as $\Delta_{SP}$ and $\Delta_{EO}$, **do not naturally support tasks other than classification**, we recognize the need to add a proper disclaimer that these recommendations are for node classification tasks. We thank you again for your suggestion on improving the quality of our work. We hope this helps address your concern.
>
> ---
>
> We thank you again for your valuable feedback on our work. With our further clarification, we believe that we have **responded to and addressed all your concerns with our point-to-point responses** — in light of this, **we hope you consider raising your score**. We again sincerely appreciate the time and efforts you've dedicated to reviewing and providing invaluable feedback to enhance the quality of our paper.

---

> > ### Comment · Reviewer_fs6n · 2024-11-25
> >
> > Thanks to the authors for a constructive rebuttal. Some points & acknowledgements:
> >
> >  * Agreed that most traditional fairness metrics only apply to node classification. I will raise my contribution score by 1.
> >  * I disagree with the claim that "the proposed evaluation strategy is novel" due to the fact that "this is the first study that has included a comprehensive set of the 10 most influential fairness-aware graph learning methods". It seems that the evaluation strategy itself has not changed from past works -- the authors unify 10 baselines under existing evaluation strategies for node classification.
> >  * re all points related to utility: yes, I missed the fact that "utility" stated colloquially throughout the paper refers to performance on the task (e.g. AUC-ROC) whereas $\Delta_{Utility}$ refers to the third fairness metric under consideration. It would help to see these distinguished more clearly in a revision.
> >
> > In general, I believe I can raise my score if I can see the changes that the authors suggested implemented in a revision. This includes qualifications re the scope of the work being limited to node classification, clarifying the distinction between "utility" and $\Delta_{Utility}$, and qualifying the conclusion about GNNs in L482.

---

> > > ### Comment · Reviewer_PqsA · 2024-11-25
> > > **fairness metrics beyond node classification**
> > >
> > > I understand that fair graph metrics extend beyond node classification, as demonstrated by works like FG²AN: Fairness-Aware Graph Generative Adversarial Networks (ECML PKDD'23).

---

> > > ### Author Response · Authors · 2024-11-25
> > > **Thank you for your kind feedback and further clarifications from authors**
> > >
> > > We sincerely appreciate your feedback, and we are glad to see that most of your concerns have been addressed. Regarding the remaining points, we would like to further clarify below:
> > >
> > > (1) We are glad to hear that your concern has been resolved!
> > >
> > > (2) We would like to clarify a misunderstanding here: the disagreement seems to originate from the definition of "evaluation strategy". In fact, we make this claim by **considering the graph learning model and dataset selection as a significant part** of the evaluation strategy. We acknowledge this phrase may lead to confusion, and we consistently used the word "Experimental Protocol" to replace this word in our manuscript to avoid further confusion. We thank you again for pointing this out.
> > >
> > > (3) We appreciate your feedback on the definition of utility, and we have revised our manuscript for better clarification.
> > >
> > > We note that **we have revised our manuscript accordingly to involve these changes**: (1) we have clarified the qualifications about the scope of the work being limited to node classification at line 193 on page 4; (2) we have clarified the distinction between "utility" and $\Delta_{Utility}$ at line 154 on page 3; (3) we have properly qualified the conclusion about GNNs in L482 (previous version) by explicitly specifying "group fairness" at lines 483-484 on page 9. **Considering the changes have been comprehensively applied, we hope you can raise your score accordingly.** We thank you again for your patience and dedicated efforts!

---

> > > > ### Comment · Reviewer_fs6n · 2024-11-25
> > > >
> > > > Thanks for the updates. I have raised my scores accordingly.

---

### Official Review · Reviewer_PmdF · 2024-11-02

**Soundness:** 2
**Presentation:** 2
**Contribution:** 2
**Rating:** 3
**Confidence:** 4

**Summary:**

The authors compare equity approaches up to 2022 through four dimensions: group fairness, individual fairness, fairness-performance trade-offs, and performance-efficiency trade-offs. They provide a more comprehensive comparison of the methods and an extensive overview of the approaches available prior to 2022.

**Strengths:**

The authors successfully reproduced the pre-2022 methods and conducted experiments using the corresponding performance indicators.

**Weaknesses:**

Defining the Balance Between Fairness Guidelines: The paper should clarify how the balance between different fairness guidelines is defined. If the balance is understood as a trade-off between fairness and performance, it is important to explain why existing related works are not utilized as baselines in the experiments [1], [2].

Unique Evaluation Metrics: The paper needs to specify whether it proposes a unique evaluation rubric for assessing fairness. If it relies solely on original metrics, it should highlight how this approach differs from current research that employs multiple baseline comparisons.

Comprehensiveness and Novelty of Fairness Perception Methods: The proposed fairness perception graph learning methods must be evaluated for their comprehensiveness and novelty to determine their potential contributions to the field. Notably, the article only includes methods up to 2022 and lacks discussion of methods from 2023 and 2024 [3], [4], [5].

Significance of Comparing Fairness Types: The significance of comparing individual and group fairness across different approaches should be addressed, especially in light of the established trade-off between these two types of fairness.

Interpretation of Figure 4: The paper should provide guidance on how to interpret the blurred trade-off represented in Figure 4. It should also discuss the insights this figure provides regarding the balance between different fairness metrics.

Interpretation of Figure 5: As fairness-aware graph learning methods, the metrics used for the baselines in Figure 5 should include fairness metrics. The absence of these metrics needs to be justified.

[1] Yuchang Zhu, et al. Fair Graph Representation Learning via Sensitive Attribute Disentanglement. ACM Web Conference 2024, 2024.

[2]  Renqiang Luo, et al. FUGNN: Harmonizing Fairness and Utility in Graph Neural Networks. KDD, 2024.

[3] Kamesh Munagala and Govind S. Sankar. Individual Fairness in Graph Decomposition. ICML, 2024.

[4] Chen Yang, et al. FairSIN: Achieving Fairness in Graph Neural Networks through Sensitive Information Neutralization. AAAI, 2024.

[5] Zhimeng Jiang, et al. Chasing Fairness in Graphs: A GNN Architecture Perspective. AAAI, 2024.

**Questions:**

How do the methods developed after 2023 perform on these benchmarks?

There has been substantial research on the trade-off between fairness and performance; why has this not been considered in the current study?

What is the contribution of comparing efficiency? When evaluating efficiency, why is performance the only metric taken into account?

---

> ### Author Response · Authors · 2024-11-23
> **Authors’ Response 1/3**
>
> We sincerely appreciate the time and efforts you've dedicated to reviewing and providing invaluable feedback to enhance the quality of this paper. We provide a point-to-point reply below for the mentioned concerns and questions.
>
> ---
>
> >  **Reviewer**: Defining the Balance Between Fairness Guidelines: The paper should clarify how the balance between different fairness guidelines is defined. If the balance is understood as a trade-off between fairness and performance, it is important to explain why existing related works are not utilized as baselines in the experiments [1], [2].
>
> **Authors**: We thank the reviewer for suggesting these recent works. We acknowledge that including newer methods like [1] and  [2] would enrich our benchmark. With that being said, in this study, we mainly **focus on the most widely used and impactful works** in the area of fairness-aware graph learning, which can directly benefit practitioners and researchers. We will also incorporate the mentioned new works in our future works and explicitly mention this limitation in our paper. We hope this helps address your concern.
>
> ---
>
> >  **Reviewer**: Unique Evaluation Metrics: The paper needs to specify whether it proposes a unique evaluation rubric for assessing fairness. If it relies solely on original metrics, it should highlight how this approach differs from current research that employs multiple baseline comparisons.
>
> **Authors**: We thank you for your insightful question. We answer the two questions below:
>
> **(1) Novel evaluation protocols:** Our study includes a series of **novel evaluation protocols**. Notably, this is the first fairness-aware graph learning work containing both shallow embedding methods and GNNs with consistent experimental settings to ensure a fair comparison. Furthermore, our study also contains new datasets (i.e., AMiner-S and AMiner-L) applicable to both group and individual fairness with the largest scale. We adopt the most commonly used fairness metrics to ensure consistency with prior works and broader applicability.
>
> **(2) How this study differs from current research that employs multiple baseline comparisons:** We note that benchmarking fairness-aware graph learning methods remains in an early stage, and related works are still rare. Qian et al. [1] took an early step to present a quantitative performance benchmark in the area of graph learning. However, they only focus on two fairness-aware GNNs, which thus blocks a broader understanding in a broader area of graph learning. Additionally, most other works lack a quantitative performance comparison, which limits an in-depth understanding of the performance of fairness-aware graph learning methods.
>
> We will also expand the corresponding discussion in Section 1 and Section 3 to enhance our paper. We hope this helps address your concern.
>
> [1] Qian, X., Guo, Z., Li, J., Mao, H., Li, B., Wang, S., & Ma, Y. (2024, August). Addressing shortcomings in fair graph learning datasets: Towards a new benchmark. In Proceedings of the 30th ACM SIGKDD Conference on Knowledge Discovery and Data Mining (pp. 5602-5612).
>
> ---
>
> >  **Reviewer**: Comprehensiveness and Novelty of Fairness Perception Methods: The proposed fairness perception graph learning methods must be evaluated for their comprehensiveness and novelty to determine their potential contributions to the field. Notably, the article only includes methods up to 2022 and lacks discussion of methods from 2023 and 2024 [3], [4], [5].
>
> **Authors**: We thank the reviewer for suggesting these recent works. We acknowledge that including newer methods like [1] and  [2] would enrich our benchmark. With that being said, in this study, we mainly **focus on the most widely used and impactful works** in the area of fairness-aware graph learning, which can directly benefit practitioners and researchers. We will also incorporate the mentioned new works in our future works. We hope this helps address your concern.

---

> ### Author Response · Authors · 2024-11-23
> **Authors’ Response 2/3**
>
> ---
>
> >  **Reviewer**: Significance of Comparing Fairness Types: The significance of comparing individual and group fairness across different approaches should be addressed, especially in light of the established trade-off between these two types of fairness.
>
> **Authors**: We thank the reviewer for the suggestion, and as suggested, we will add a detailed discussion on the significance of comparing individual and group fairness across different approaches. Here we provide an introduction below.
>
> We note that **group fairness and individual fairness address fairness at different levels**: group fairness ensures non-discrimination across demographic groups, while individual fairness ensures similar treatment for similar individuals. The reviewer’s concern highlights the importance of explicitly analyzing and understanding the trade-off between these two fairness notions, as improving one often compromises the other.
>
> Additionally, the balance between individual and group fairness is not merely a theoretical concern but also **a practical need**, especially in high-stakes applications like credit lending and healthcare, where conflicting fairness objectives may arise. **We agree with the reviewer that highlighting this relevance in real-world contexts will strengthen the paper’s argument.**
>
> We hope the detailed discussion above helps address your concern.
>
> ---
>
> >  **Reviewer**: Interpretation of Figure 4: The paper should provide guidance on how to interpret the blurred trade-off represented in Figure 4. It should also discuss the insights this figure provides regarding the balance between different fairness metrics.
>
> **Authors**: We appreciate the reviewer’s feedback regarding the interpretation of Figure 4 and the trade-offs represented within it. Figure 4 demonstrates the rankings of different fairness-aware graph learning methods based on three fairness metrics ($\Delta_{SP}$, $\Delta_{EO}$, and $\Delta_{Utility}$) across all datasets. **These metrics collectively highlight the tension between achieving statistical parity, equal opportunity, and minimizing utility differences among demographic groups.**
>
> Specifically, shallow embedding methods (e.g., FairWalk and CrossWalk) tend to perform well on group fairness metrics ($\Delta_{SP}$ and $\Delta_{EO}$) due to their simplicity and lack of reliance on node attributes, which can often encode biases. Conversely, GNN-based methods, such as FairGNN and FairEdit, excel in ΔUtilityΔUtility, likely benefiting from their ability to utilize node attribute information for subgroup performance consistency. However, these methods generally show less balance across all three metrics, indicating that fairness-aware graph learning methods inherently struggle to optimize multiple fairness objectives simultaneously. This highlights the importance of method selection based on the specific fairness goals of a given application.
>
> We will revise the manuscript to include these insights and provide more detailed guidance for practitioners in interpreting the trade-offs visualized in Figure 4. We hope the detailed explanation above helps address your concern.
>
> ---
>
> >  **Reviewer**: Interpretation of Figure 5: As fairness-aware graph learning methods, the metrics used for the baselines in Figure 5 should include fairness metrics. The absence of these metrics needs to be justified.
>
> **Authors**: We thank you for the insightful question. We note that improving fairness typically leads to heavier computational burdens, and this can be a serious concern when the ultimate goal is to deploy in real-world applications. Therefore, **our comparison of efficiency stands out from existing studies by providing a unique way to evaluate the cost of these methods for real-world applications**. The reason for only considering performance is that how much performance is compromised will be the other key concern for real-world deployment. We hope this helps address your concern.

---

> ### Author Response · Authors · 2024-11-23
> **Authors’ Response 3/3**
>
> ---
>
> >  **Reviewer**: How do the methods developed after 2023 perform on these benchmarks?
>
> **Authors**: We thank the reviewer for suggesting these recent works. We acknowledge that also including methods after 2023 would enrich our benchmark. With that being said, in this study, we mainly **focus on the most widely used and impactful works** in the area of fairness-aware graph learning, which can directly benefit practitioners and researchers. We will also incorporate the mentioned new works in our future works. We hope this helps address your concern.
>
> ---
>
> >  **Reviewer**: There has been substantial research on the trade-off between fairness and performance; why has this not been considered in the current study?
>
> **Authors**: Thank you for raising this point. While existing studies on the fairness-performance trade-off offer valuable insights, they often focus on **specific fairness notions** or **inconsistent experimental settings**, limiting their generalizability. Additionally, **inconsistent evaluation protocols** hinder comprehensive understanding. In contrast, our work introduces a **systematic benchmark** of ten representative fairness-aware graph learning methods, evaluated under a **consistent protocol** across diverse metrics (e.g., utility, group fairness, individual fairness) and real-world datasets. This provides a **global perspective** that enables direct comparison and offers practical guidance for applications, making our contribution **unique and complementary** to prior studies. We hope this helps address your concern.
>
> ---
>
> >  **Reviewer**: What is the contribution of comparing efficiency? When evaluating efficiency, why is performance the only metric taken into account?
>
> **Authors**: We thank you for the insightful question. We note that **lacking efficiency comparison is another issue that has long been ignored by other existing studies**. Specifically, improving fairness typically leads to heavier computational burdens, and this can be a serious concern when the ultimate goal is to deploy in real-world applications. Therefore, **our comparison of efficiency stands out from existing studies by providing a unique way to evaluate the cost of these methods for real-world applications**. The reason for only considering performance is that how much performance is compromised will be the other key concern for real-world deployment. We hope this helps address your concern.
>
> ---
>
> We thank you again for your valuable feedback on our work. With our further clarification, we believe that we have **responded to and addressed all your concerns with our point-to-point responses** — in light of this, **we hope you consider raising your score**.  We again sincerely appreciate the time and efforts you've dedicated to reviewing and providing invaluable feedback to enhance the quality of our paper.

---

> ### Author Response · Authors · 2024-11-25
> **A Kind Reminder**
>
> Dear Reviewer PmdF,
>
> Thank you for your valuable feedback on our work. We have prepared a thorough response to address your concerns. We believe that we have responded to and addressed all your concerns with our responses — in light of this, **we hope you consider raising your score**.
>
> Notably, given that we are approaching the deadline for the rebuttal phase, we hope we can receive your feedback soon. We thank you again for your efforts and suggestions!

---

> > ### Comment · Reviewer_PmdF · 2024-11-26
> >
> > 1. Avoidance of Baseline Comparisons: The authors fail to adequately address concerns about the lack of meaningful and relevant baseline comparisons. Instead of providing sufficient evidence to support their claims, they avoid comparing their method to state-of-the-art approaches, leaving the evaluation incomplete and unconvincing.
> >
> > 2. Undefined "the most widely used and impactful works": The claim of broader applicability is neither defined nor substantiated. Other fairness methods can also operate across multiple datasets, yet the authors neglect to clarify why their approach is more widely applicable or to conduct direct comparisons with closely related fairness techniques.
> >
> > 3. Performance Deficiency: The proposed method demonstrates significant gaps in both utility and fairness metrics compared to stronger existing methods.  This discrepancy undermines the claim of improved performance and calls into question the practical advantages of the proposed framework.
> >
> > In its current form, the submission lacks sufficient experimental support and theoretical rigor to justify its claims, making it unsuitable for acceptance.

---

### Official Review · Reviewer_PqsA · 2024-11-03

**Soundness:** 2
**Presentation:** 2
**Contribution:** 1
**Rating:** 3
**Confidence:** 5

**Summary:**

In this paper, the authors aim to establish a benchmark for fairness-aware graph learning to guide practical applications by evaluating the trade-off between fairness and performance of multiple graph learning methods. To this end, the authors propose a set of evaluation protocols covering group fairness and individual fairness metrics, summarize and test the fairness and performance of ten representative fairness graph learning methods on seven real-world datasets, provide systematic experimental results and analyses demonstrating their trade-offs between fairness and utility, as well as constructing a reliable and quantitative basis for a benchmark for fairness graph learning.

**Strengths:**

●	The authors address an interesting and timely topic that is absolutely relevant to ICLR.

●	The authors tested these methods on seven real-world datasets, including datasets from different domains such as social networks and finance.

**Weaknesses:**

●	The novelty of this paper is limited, as the systematic evaluation approach proposed does not significantly improve upon existing methods. The authors primarily use visual comparisons, i.e., plotting accuracy and fairness or presenting them separately in tables or bar charts. This approach is still unable to effectively illustrate the trade-off between performance and fairness for fair GNN.
●	The comparison of fair graph methods are some of the early works. The authors omitting newer approaches such as Graphair [1] and FairSAD [2], which should be included for a more comprehensive comparison.
[1] Ling, Hongyi, et al. "Learning fair graph representations via automated data augmentations." International Conference on Learning Representations (ICLR). 2023.
[2] Zhu, Yuchang, et al. "Fair Graph Representation Learning via Sensitive Attribute Disentanglement." Proceedings of the ACM on Web Conference 2024. 2024.
●	The discussion of experimental results lacks depth and context. The authors mostly describe the numerical scores shown in the result table without interpreting their significance. For example, in RO2, three individual fairness metrics are used to assess model fairness, yet the authors do not discuss the rationale for choosing multiple metrics, the potential conflicts among them, or how to interpret whether the metric values indicate good or poor performance.

**Questions:**

please refer the weakness.

---

> ### Author Response · Authors · 2024-11-23
> **Authors’ Response 1/2**
>
> We sincerely appreciate the time and efforts you've dedicated to reviewing and providing invaluable feedback to enhance the quality of this paper. We provide a point-to-point reply below for the mentioned concerns and questions.
>
> ---
>
> >  **Reviewer**: ● The novelty of this paper is limited, as the systematic evaluation approach proposed does not significantly improve upon existing methods. The authors primarily use visual comparisons, i.e., plotting accuracy and fairness or presenting them separately in tables or bar charts. This approach is still unable to effectively illustrate the trade-off between performance and fairness for fair GNN.
>
> **Authors**: We thank the reviewer for the feedback. We would like to clarify a misunderstanding that **novelty should not solely be judged by whether improved performance is reported**. We believe that novelty also lies in the new insights a research work can bring to the research community and the potential new impact it can have. To this end, we would like to emphasize the novelty of our work as: (1) **Systematic Evaluation Protocol**: To the best of our knowledge, this is the first study introducing a comprehensive protocol exploring multiple popular perspectives in evaluating fairness-aware graph learning methods; (2) **Comprehensive Benchmark**: This is the first-of-its-kind benchmark that integrates both commonly used and newly constructed datasets to explore the strengths and weaknesses of both GNNs and shallow embedding methods on fairness, utility, and efficiency; (3) **Practical Guidance**: This study also offers novel and actionable insights for practitioners, bridging the gap between the current advances in fairness-aware graph learning and real-world applications.
>
> Additionally, we recognize that while visual comparisons are intuitive, they may not fully capture the trade-offs between performance and fairness. However, we would also like to note that all corresponding quantitative results have also been comprehensively reported in the appendix. To address the mentioned concern, we will enhance Section 4.3 by adding quantitative analyses such as Pareto efficiency evaluations and trade-off curves, explicitly illustrating the balance between utility and fairness metrics for each method. We hope this helps address your concern.
>
> ---
>
> >  **Reviewer**: ● The comparison of fair graph methods are some of the early works. The authors omitting newer approaches such as Graphair [1] and FairSAD [2], which should be included for a more comprehensive comparison. [1] Ling, Hongyi, et al. "Learning fair graph representations via automated data augmentations." International Conference on Learning Representations (ICLR). 2023. [2] Zhu, Yuchang, et al. "Fair Graph Representation Learning via Sensitive Attribute Disentanglement." Proceedings of the ACM on Web Conference 2024. 2024.
>
> **Authors**: We thank the reviewer for suggesting these recent works. We acknowledge that including newer methods like Graphair [1] and FairSAD [2] would enrich our benchmark. With that being said, in this study, we mainly **focus on the most widely used and impactful works** in the area of fairness-aware graph learning, which can directly benefit practitioners and researchers. We will also incorporate the mentioned new works in our future works and explicitly mention this limitation in our paper. We hope this helps address your concern.

---

> ### Author Response · Authors · 2024-11-23
> **Authors’ Response 2/2**
>
> ---
>
> >  **Reviewer**: ● The discussion of experimental results lacks depth and context. The authors mostly describe the numerical scores shown in the result table without interpreting their significance. For example, in RO2, three individual fairness metrics are used to assess model fairness, yet the authors do not discuss the rationale for choosing multiple metrics, the potential conflicts among them, or how to interpret whether the metric values indicate good or poor performance.
>
> **Authors**: We thank the reviewer for the suggestion. We agree with the reviewer that it will be helpful to discuss the rationale for choosing multiple metrics, the potential conflicts among them, and how to interpret whether the metric values indicate good or poor performance. In fact, **we have briefly mentioned them in Section 2**. Below we provide a more detailed discussion.
>
> **(1) The rationale for choosing multiple metrics**: Since different types of fairness can be favored by different real-world applications, the paper uses multiple individual fairness metrics to capture different aspects of fairness. $B_{\text {Lipschitz }}$ evaluates how similar outputs can be received by individuals with similar profiles; $N D C G @ k$ measures how well the ranking consistency of input can be preserved in the output space; $G D I F$ evaluates the equity of individual fairness across demographic subgroups. Together, these metrics provide a comprehensive evaluation of individual fairness, aligning with those most widely used criteria.
>
> **(2) Potential Conflicts Among Metrics**: Conflicts can arise when optimizing one metric negatively impacts another. For example, improving GDIF may reduce $B_{\text {Lipschitz }}$ by enforcing subgroup-level equity at the cost of individual consistency. Similarly, prioritizing $N D C G @ k$ might compromise $B_{\text {Lipschitz }}$ by focusing on relative rankings over exact proximities. An evidence is that a model can hardly do well on all three metrics, making versatility on individual fairness difficult to achieve.
>
> **(3) Interpretation of Metric Values**: Low $B_{\text {Lipschitz }}$ and $G D I F$ values, combined with high $N D C G @ k$, indicate good fairness, while high values of $B_{\text {Lipschitz }}$ or $G D I F$ suggest poor performance. We further note that metrics should be interpreted contextually, balancing trade-offs based on application priorities.
>
> Accordingly, based on the discussion above, we will expand Section 4.2 to provide detailed interpretations of the individual fairness metrics, explaining their rationale and the potential conflicts among them. Additionally, we will discuss how these metrics align with fairness objectives and provide guidance on interpreting the results. We hope this helps address your concern.
>
> ---
>
> We thank you again for your valuable feedback on our work. With our further clarification, we believe that we have **responded to and addressed all your concerns with our point-to-point responses** — in light of this, **we hope you consider raising your score**.  We again sincerely appreciate the time and efforts you've dedicated to reviewing and providing invaluable feedback to enhance the quality of our paper.

---

> ### Author Response · Authors · 2024-11-25
> **A Kind Reminder**
>
> Dear Reviewer PqsA,
>
> Thank you for your valuable feedback on our work. We have prepared a thorough response to address your concerns. We believe that we have responded to and addressed all your concerns with our responses — in light of this, **we hope you consider raising your score**.
>
> Notably, given that we are approaching the deadline for the rebuttal phase, we hope we can receive your feedback soon. We thank you again for your efforts and suggestions!

---

### Official Review · Reviewer_Zqpq · 2024-11-03

**Soundness:** 3
**Presentation:** 3
**Contribution:** 2
**Rating:** 5
**Confidence:** 4

**Summary:**

This work addresses the pressing issue of fairness in graph learning by providing a comprehensive evaluation of ten fairness-aware graph learning methods. The authors propose an experimental setup to evaluate these methods across seven real-world datasets, focusing on group fairness, individual fairness, the balance between fairness and utility, and computational efficiency. Their systematic approach aims to offer guidance to practitioners for selecting appropriate fairness-aware methods in real-world applications.

**Strengths:**

1. Originality: Although the motivation and the idea to do this kind of study have been attempted in the past, as pointed out by the authors, they also make a clear distinction of how their approach is different.

2. Quality: The empirical evaluation is extensive, spanning several datasets and including multiple fairness and utility metrics. Common checkmarks for benchmark works like hyperparameter tuning, reporting across multiple runs, etc., are taken care of.

3. Clarity: Key objectives and methods are introduced clearly, and the premise is set clearly without any confusion on the objective of the work. The work is very easy to read and follow, and the figures and the metrics reported are well explained.

4. Significance: Fairness in machine learning on graphs is a growing area of importance, and this benchmark offers a much-needed reference point for comparing fairness-aware graph methods. While there are more definitions of fairness in this domain, this work covers the most widely used settings.

**Weaknesses:**

W1. The paper implies that this benchmark is a comprehensive evaluation of fairness-aware methods, yet it evaluates only a subset of fairness criteria. There are prior works that cover more inherent fairness issues in GNNs that do not depend upon the node characteristics [1] [2], and the evaluation of such methods would have also been a great addition.

W2. While the benchmark includes a selection of ten methods, the paper does not sufficiently discuss recent advancements in fairness-aware graph learning, such as methods that address intersectional biases or temporal fairness. This limits the relevance of the benchmark for contemporary applications and makes it appear somewhat outdated in the rapidly evolving fairness landscape.

W3. Experimental details are insufficiently described. Important aspects such as hyperparameter search ranges, validation criteria, and dataset preprocessing steps are either missing or vaguely specified. While Figure 3 conveys some parts of it, the exact ranges specified and the final hyperparameters used would be helpful for reproducibility.

W4. The benchmarking focuses mainly on fairness-aware graph neural networks (GNNs) and shallow embeddings but lacks methods that might employ fairness-enhancing architectures or hybrid models. Expanding this range would make the benchmark more relevant and impactful.

W5. The benchmark briefly touches on computational efficiency but does not thoroughly investigate the scalability of these fairness-aware methods on large-scale graph datasets. Given that scalability is a crucial consideration for real-world applications, particularly in social networks or knowledge graphs, this is a notable omission.

W6. While the findings validate existing assumptions in the field, they don't offer novel insights beyond what practitioners would already understand through experience.

[1] Liu, Zemin, Trung-Kien Nguyen, and Yuan Fang. "On generalized degree fairness in graph neural networks." Proceedings of the AAAI Conference on Artificial Intelligence. Vol. 37. No. 4. 2023.

[2] Arun, Arvindh, et al. "CAFIN: Centrality Aware Fairness Inducing IN-Processing for Unsupervised Representation Learning on Graphs." ECAI 2023.

**Questions:**

Q1. Can the authors clarify the hyperparameter tuning process? Specifically, were grid search ranges adjusted for each dataset, and how were optimal settings chosen? What were the search ranges, and how were they decided upon?

Q2. The dataset selection includes only node classification tasks. Can the authors comment on the applicability of this benchmark for link prediction or other graph-based tasks?

Q3. More realistic scenarios that often have hybrid fairness objectives have not been tested; what would be the performance of these methods in those settings?

Q4. How does the benchmark account for potential variations in fairness definitions across different domains, especially when the impact of bias may differ (e.g., financial vs. social networks)?

Q5. It would be interesting to see the GNN methods running without (or with random) node attributes since the claim that the "absence of bias brought by node attributes" is mentioned in two places in the text without any supporting evidence.

Q6. The confidence intervals in Figure 2 overlap a lot, which is why I'm unsure about the statistical validity of the claims. Can the authors verify the significance of the results presented?

---

> ### Author Response · Authors · 2024-11-23
> **Authors’ Response 1/4**
>
> We sincerely appreciate the time and efforts you've dedicated to reviewing and providing invaluable feedback to enhance the quality of this paper. We provide a point-to-point reply below for the mentioned concerns and questions.
>
> ---
>
> >  **Reviewer**: W1. The paper implies that this benchmark is a comprehensive evaluation of fairness-aware methods, yet it evaluates only a subset of fairness criteria. There are prior works that cover more inherent fairness issues in GNNs that do not depend upon the node characteristics [1] [2], and the evaluation of such methods would have also been a great addition.
>
> **Authors**: We thank the reviewer for this constructive feedback. We agree that expanding the benchmark to include additional fairness criteria, such as fairness notions independent of node characteristics, can provide a more comprehensive evaluation. While our current scope emphasizes well-established fairness notions, we will include a discussion in the related work section to highlight these alternative approaches and identify them as promising directions for future works. We hope this helps address your concern.
>
> ---
>
> >  **Reviewer**: W2. While the benchmark includes a selection of ten methods, the paper does not sufficiently discuss recent advancements in fairness-aware graph learning, such as methods that address intersectional biases or temporal fairness. This limits the relevance of the benchmark for contemporary applications and makes it appear somewhat outdated in the rapidly evolving fairness landscape.
>
> **Authors**: We thank the reviewer for pointing this out. We acknowledge that recent advancements, such as addressing intersectional and temporal fairness, are underexplored in our benchmark. However, we note that certain fairness metrics that are not commonly used can be **incompatible with the popular graph learning methods**. For example, all GNN-based methods adopted in this paper cannot take time-varying signals as input and will not be compatible with being evaluated with temporal fairness metrics, making the evaluation goes beyond the main focus of our work.
>
> Additionally, we argue that despite the rapidly evolving landscape of fairness-aware graph learning, we argue that this study remains highly timely noticing three key novelties below.
>
> 1. **Systematic Evaluation Protocol**: This is the first study to introduce a comprehensive protocol exploring multiple popular perspectives in evaluating fairness-aware graph learning methods.
> 2. **Comprehensive Benchmark**: We present the first-of-its-kind benchmark that integrates both commonly used and newly constructed datasets, enabling a thorough exploration of the strengths and weaknesses of both GNNs and shallow embedding methods with respect to fairness, utility, and efficiency.
> 3. **Practical Guidance**: Our study offers novel and actionable insights for practitioners, bridging the gap between current advances in fairness-aware graph learning and real-world applications.
>
> We hope the detailed explanation above helps address your concern.
>
> ---
>
> >  **Reviewer**: W3. Experimental details are insufficiently described. Important aspects such as hyperparameter search ranges, validation criteria, and dataset preprocessing steps are either missing or vaguely specified. While Figure 3 conveys some parts of it, the exact ranges specified and the final hyperparameters used would be helpful for reproducibility.
>
> **Authors**: We thank the reviewer for this observation. We agree that clear experimental details are crucial for our study. We will provide a comprehensive table detailing hyperparameter search ranges, validation criteria, and preprocessing steps for each dataset and method in the appendix of our next version. Notably, we will also make the source code publically available for reproducibility purposes. We hope this helps address your concern.

---

> ### Author Response · Authors · 2024-11-23
> **Authors’ Response 2/4**
>
> ---
>
> >  **Reviewer**: W4. The benchmarking focuses mainly on fairness-aware graph neural networks (GNNs) and shallow embeddings but lacks methods that might employ fairness-enhancing architectures or hybrid models. Expanding this range would make the benchmark more relevant and impactful.
>
> **Authors**: We thank the reviewer for this suggestion. We would like to clarify **a misunderstanding that most of the adopted methods**, including the GNN-based and shallow embedding ones, **are based on either fairness-enhancing architectures or hybrid design**. For example, the famous FairWalk and CrossWalk are in fact integrated with a hybrid design, enforcing fairness and utility (i.e., quality of the learned representations) at the same time. As another example, multiple GNN-based approaches, such as FairGNN and FairVGNN, actually enjoy carefully designed architectures that help improve fairness. Meanwhile, there are also approaches that directly modify the architectures of GNNs themselves, such as EDITS.
>
> Additionally, the reason to choose the two types of methods for comparison is that they are two mainstreams of graph learning in this line of research. Benchmarking these widely used models can directly help researchers and practitioners and benefit further advancements in this field. We hope this helps address your concern.
>
> ---
>
> >  **Reviewer**: W5. The benchmark briefly touches on computational efficiency but does not thoroughly investigate the scalability of these fairness-aware methods on large-scale graph datasets. Given that scalability is a crucial consideration for real-world applications, particularly in social networks or knowledge graphs, this is a notable omission.
>
> **Authors**: We thank you for mentioning this concern. We would like to highlight a misunderstanding here: **only a limited number of graph datasets are available for fairness-aware learning tasks**. For example, most group fairness metrics require that sensitive attributes (e.g., gender or race) should be explicitly contained in the dataset to perform calculations. However, the commonly used large-scale graph datasets, such as the well-known OGB series, do not have such information and cannot be adopted for evaluation.
>
> We would like to highlight that we have already involved the **two largest social network datasets** in the field of fairness-aware graph learning, Pokec-z and Pokec-n, in our paper. Additionally, we also collected two even larger citation networks, AMiner-S and AMiner-L, from the well-known AMiner platform for our benchmark. Therefore, we argue that to the best of our knowledge, this study serves as the fairness-aware graph learning benchmark **involving the largest graph datasets applicable to this line of research**. We hope our detailed explanation above helps address your concern.
>
> ---
>
> >  **Reviewer**: W6. While the findings validate existing assumptions in the field, they don't offer novel insights beyond what practitioners would already understand through experience.
>
> **Authors**: We thank the reviewer for the feedback. We would like to highlight that there are a series of key insights outlined in Section 4 that have either not been mentioned in other works or challenged traditional opinions. Below we introduce two representative insights that have **rarely been reported in other studies** and **can potentially be impactful** in future advancements.
>
> (1) **Different from the common belief that fairness-aware GNNs can achieve better fairness**, they usually achieve a better balance between utility and fairness. In fact, shallow fairness-aware approaches such as FairWalk and CrossWalk can achieve the best performances on group fairness metrics, which due to their focus on rebalancing graph structure rather than using node attributes (which often carry bias). Meanwhile, improving the group fairness levels **does not necessarily mean compromise on utility**.
>
> (2) GUIDE, which takes both group and individual fairness into consideration, achieves the best balance between different individual fairness criteria — the compositional design of its objective allowed it to balance different individual fairness metrics effectively. **Different from group fairness**, individual fairness typically leads to stronger compromise on utility.
>
> We will also expand Section 4 on those novel insights that directly benefit the practitioners and researchers. We hope the detailed explanation above helps address your concern.

---

> ### Author Response · Authors · 2024-11-23
> **Authors’ Response 3/4**
>
> ---
>
> >  **Reviewer**: Q1. Can the authors clarify the hyperparameter tuning process? Specifically, were grid search ranges adjusted for each dataset, and how were optimal settings chosen? What were the search ranges, and how were they decided upon?
>
> **Authors**: We thank the reviewer for this question. Hyperparameter tuning was performed using grid search with dataset-specific ranges. The ranges were determined based on the associated original studies (for commonly used datasets) and our empirical experiments (for datasets that have not been used in original studies). The optimal settings were selected based on the lowest validation loss. We will also expand the discussion about more details, e.g., tuning ranges and selecting criteria above, in the appendix for clarity. We hope this helps address your concern.
>
> ---
>
> >  **Reviewer**: Q2. The dataset selection includes only node classification tasks. Can the authors comment on the applicability of this benchmark for link prediction or other graph-based tasks?
>
> **Authors**: We thank the reviewer for raising this question. We note that **most traditional fairness metrics**, such as $\Delta_{SP}$ and $\Delta_{EO}$, **do not naturally support tasks other than classification**. Such compatibility has been widely acknowledged by this line of research — as a consequence, we choose the most widely adopted node classification task to conduct our benchmark to ensure broader applicability to both research and applications. We agree with the reviewer that having a benchmark that can be used for various tasks is attractive, while this is a difficult task since **it requires fundamental changes to the commonly used fairness definition and metrics**. This has gone beyond the main goal of this paper, and we will leave it as future work. We hope this helps address your concern.
>
> ---
>
> >  **Reviewer**: Q3. More realistic scenarios that often have hybrid fairness objectives have not been tested; what would be the performance of these methods in those settings?
>
> **Authors**: We thank the reviewer for this observation. Testing hybrid fairness objectives is indeed valuable and aligns with real-world requirements. While these scenarios were not included in the current study, we will mention this limitation and outline a future plan to extend the benchmark to incorporate hybrid fairness objectives.
>
> ---
>
> >  **Reviewer**: Q4. How does the benchmark account for potential variations in fairness definitions across different domains, especially when the impact of bias may differ (e.g., financial vs. social networks)?
>
> **Authors**: We thank the reviewer for this insightful feedback. Our benchmark evaluates fairness **using widely accepted definitions**, which aligns with the desiderata of fairness in most applications (e.g., $\Delta_{SP}$ and $\Delta_{EO}$ acknowledge positive predictions represent favorable results in loan applications or other financial services). However, the variations in fairness definitions across different domains **has not been thoroughly discussed in this line of research**. In fact, this is a difficult problem that requires interdisciplinary efforts. We will carefully expand Section 3.1 to discuss how domain-specific fairness definitions could influence outcomes and we plan to propose tailored evaluations for distinct domains in future work. We hope this helps address your concern.
>
> ---
>
> >  **Reviewer**: Q5. It would be interesting to see the GNN methods running without (or with random) node attributes since the claim that the "absence of bias brought by node attributes" is mentioned in two places in the text without any supporting evidence.
>
> **Authors**: We thank you for your insightful question. In fact, we have involved EDITS in our benchmark, and the goal of this model is to **first debias the node attributes** and then **use traditional GNNs** to test the performance. We are able to observe **significant fairness improvements** in most cases, and this verifies that the bias encoded in the node attributes can be a significant source of bias in the output prediction. Considering that using random node attributes can severely jeopardize the performance of GNNs and may lose practical significance, the observations on EDITS will be ideal evidence to answer your insightful question. We hope this helps address your concern.

---

> ### Author Response · Authors · 2024-11-23
> **Authors’ Response 4/4**
>
> ---
>
> >  **Reviewer**: Q6. The confidence intervals in Figure 2 overlap a lot, which is why I'm unsure about the statistical validity of the claims. Can the authors verify the significance of the results presented?
>
> **Authors**: We thank you for pointing this out. We note that certain variances can be large in Figure 2 since the calculation is based on discrete ranking orders, and we acknowledge that not all rankings in Figure 2 show statistical significance. However, the models we have claimed superiority, such as $\Delta_{SP}$ on CrossWalk and FairWalk, bear **significantly lower variances** and enjoy **solid statistical significance compared with most other models**. We will expand the discussion in Section 4.1 to involve quantitative results and significance tests and we hope this helps address your concern.
>
> ---
>
> We thank you again for your valuable feedback on our work. With our further clarification, we believe that we have **responded to and addressed all your concerns with our point-to-point responses** — in light of this, **we hope you consider raising your score**.  We again sincerely appreciate the time and efforts you've dedicated to reviewing and providing invaluable feedback to enhance the quality of our paper.

---

> ### Author Response · Authors · 2024-11-25
> **A Kind Reminder**
>
> Dear Reviewer Zqpq,
>
> Thank you for your valuable feedback on our work. We have prepared a thorough response to address your concerns. We believe that we have responded to and addressed all your concerns with our responses — in light of this, **we hope you consider raising your score**.
>
> Notably, given that we are approaching the deadline for the rebuttal phase, we hope we can receive your feedback soon. We thank you again for your efforts and suggestions!

---

> > ### Comment · Reviewer_Zqpq · 2024-11-25
> > **Clarification regarding the update**
> >
> > Thank you for the clarifications. Are these proposed changes implemented in the current version? If so, I request the authors to point out the specific places in the paper where changes addressing my concerns have been made.
> >
> > > "Considering that using random node attributes can severely jeopardize the performance of GNNs and may lose practical significance, the observations on EDITS will be ideal evidence to answer your insightful question."
> >
> > I respectfully disagree completely with the authors on this statement [1]. This, again, sounds like an empty claim without any experimental backing. Despite how the results turn out, I would still like to see the experimental results with all the GNN baselines with random node attributes for me to derive anything conclusive.
> >
> > I will wait for this clarification and the details about the exact changes made addressing my concerns before re-evaluating my score.
> >
> > [1] Abboud, R., Ceylan, İ. İ., Grohe, M., & Lukasiewicz, T. (8 2021). The Surprising Power of Graph Neural Networks with Random Node Initialization. In Z.-H. Zhou (Ed.), Proceedings of the Thirtieth International Joint Conference on Artificial Intelligence, IJCAI-21 (pp. 2112–2118). doi:10.24963/ijcai.2021/291

---

> > > ### Author Response · Authors · 2024-11-26
> > >
> > > We thank you for your feedback, and we will be glad to to point out the changes we had made to our manuscripts to address your concerns. Considering that most concern comes from the clarifications regarding w1 and w6, we have made changes to line 336 - 341 and 356 to further clarify w6 and line 521 - 522 to further clarify w1. Regarding the details of how the hyper-parameters are tuned (q1), we will release together with our open-source code upon acceptance. Additionally, we have further provided detailed quantitative results in our appendix for significance test (q6).
> > >
> > > Regarding your last concern, we would like to review that your concern is:
> > >
> > > >  if "graph structure + node attributes -> biased predictions" and node attributes contain bias, then will remove node attributes lead to less bias?
> > >
> > > Since simply removing node attributes is naturally not feasible for GNNs, we address your concern by the evidence provided by EDITS that:
> > >
> > > > Since "graph structure + less biased node attributes -> less biased predictions", your intuition is correct and aligns with the practical evidence.
> > >
> > > Additionally, we note that we have also performed experiments on three representative datasets that are widely used by this line of research works with carefully parameter tuning. We have followed your request to show "experimental results with all the GNN baselines with random node attributes" for you below. Compared with the results in Table 3 and the corresponding results in Table 1 in our appendix, we are able to find that the utility (measured with ACC and AUCROC) has droped by a large margin for most cases, and the fairness levels have also been improved at the same time, which aligns with your intuition and our claim as well. We believe that we have responded to and addressed all your concerns with our responses — in light of this, **we hope you consider raising your score**.
> > >
> > > ### German Dataset
> > >
> > > | Method   | ACC   | AUCROC | F1     | SP     | EO     |
> > > | -------- | ----- | ------ | ------ | ------ | ------ |
> > > | EDITS    | 0.488 | 0.5162 | 0.488  | 0.1020 | 0.0777 |
> > > | FairGNN  | 0.564 | 0.5857 | 0.564  | 0.2578 | 0.2300 |
> > > | FairVGNN | 0.640 | 0.5906 | 0.7429 | 0.0818 | 0.0945 |
> > > | FairWalk | 0.664 | 0.5352 | 0.664  | 0.0412 | 0.0788 |
> > > | GNN      | 0.688 | 0.6933 | 0.688  | 0.2219 | 0.1334 |
> > > | NIFTY    | 0.468 | 0.4943 | 0.468  | 0.2661 | 0.2626 |
> > >
> > > ---
> > >
> > > ### Recidivism Dataset
> > >
> > > | Method   | ACC    | AUCROC | F1     | SP     | EO     |
> > > | -------- | ------ | ------ | ------ | ------ | ------ |
> > > | EDITS    | 0.5202 | 0.5070 | 0.5202 | 0.0135 | 0.0170 |
> > > | FairGNN  | 0.5622 | 0.5158 | 0.5622 | 0.0233 | 0.0197 |
> > > | FairVGNN | 0.5514 | 0.5380 | 0.4347 | 0.0060 | 0.0231 |
> > > | FairWalk | 0.8856 | 0.8575 | 0.8856 | 0.0575 | 0.0367 |
> > > | GNN      | 0.8909 | 0.8716 | 0.8909 | 0.0799 | 0.0579 |
> > > | NIFTY    | 0.4914 | 0.4998 | 0.4914 | 0.0083 | 0.0145 |
> > >
> > > ---
> > >
> > > ### Credit Dataset
> > >
> > > | Method   | ACC    | AUCROC | F1     | SP     | EO     |
> > > | -------- | ------ | ------ | ------ | ------ | ------ |
> > > | EDITS    | 0.5205 | 0.5064 | 0.5205 | 0.0071 | 0.0315 |
> > > | FairGNN  | 0.6784 | 0.5014 | 0.6784 | 0.0156 | 0.0063 |
> > > | FairVGNN | 0.6968 | 0.4937 | 0.8149 | 0.0172 | 0.0344 |
> > > | FairWalk | 0.7495 | 0.5403 | 0.7495 | 0.0487 | 0.0402 |
> > > | GNN      | 0.4969 | 0.4936 | 0.4969 | 0.0770 | 0.0681 |
> > > | NIFTY    | 0.5381 | 0.4935 | 0.5381 | 0.0034 | 0.0011 |

---

### Official Review · Reviewer_NAeS · 2024-11-04

**Soundness:** 3
**Presentation:** 4
**Contribution:** 2
**Rating:** 5
**Confidence:** 3

**Summary:**

This paper provided a comprehensive protocol to evaluate the performance of fairness-aware graph learning methods. Extensive experiments on seven real-world attributed graph datasets with ten fairness-aware graph learning methods were conducted to induce in-depth analysis for the benchmark results.

**Strengths:**

1. This paper provided a comprehensive benchmark over ten very recent fairness-aware graph learning methods, including both group fairness and individual fairness. A clear timeline and categorization were provided to help the presentation of the whole study.
2. Extensive experiments on seven datasets and ten methods were conducted and were well organized to answer the four research questions. Various metrics in utility, group fairness and individual fairness were compared.
3. In Section 4, the results in the tables were further partially compared in figures to discuss the limitations and strengths of different methods, which validated the findings of the authors.
4. Practical guidance was provided for users to help choose the most appropriate fairness-aware graph learning methods.

**Weaknesses:**

1. The discussion mostly focused on stating the observations that different methods have different performances on different metrics. However, a more in-depth discussion of why these methods have different advantages could be added. E.g., how do different objective functions in the methods benefit certain fairness notions?
2. For a benchmark study of fair graph learning, how do you choose the datasets for testing? What is the group density in the graphs? Are the groups balanced or not? These details could be clarified to justify the diverse choice of datasets for evaluation.
3. What are the connections and the comparison within different group fairness metrics, within different individual fairness metrics, and between them? Can you provide more explanations in Section 4.3?
4. Although the authors studied efficiency in Section 4.4, it is not well discussed as a factor in their practical guidance.

In all, despite the extensive experiments, more in-depth analysis could be added instead of simply presenting the results. Please also see the minor weaknesses in the questions.

**Questions:**

1. I did not find the newly constructed datasets AMiner-S and AMinder-L in the supplemental materials, which I think may count for part of the contributions of this work. Am I missing something?
2. In lines 334-335 you mentioned "neither DeepWalk nor GNN yields top-ranked performance under utility". Where does this statement come from? DeepWalk achieves an even worse ranking in utility than fairness.
3. In Figure 3, what do dots of different colors stand for?

---

> ### Author Response · Authors · 2024-11-23
> **Authors’ Response 1/2**
>
> We sincerely appreciate the time and efforts you've dedicated to reviewing and providing invaluable feedback to enhance the quality of this paper. We provide a point-to-point reply below for the mentioned concerns and questions.
>
> ---
>
> >  **Reviewer**: The discussion mostly focused on stating the observations that different methods have different performances on different metrics. However, a more in-depth discussion of why these methods have different advantages could be added. E.g., how do different objective functions in the methods benefit certain fairness notions?
>
> **Authors**: We thank the reviewer for the constructive feedback, and we agree with the reviewer that a more in-depth discussion will make this paper even more interesting. To achieve this goal, we have **carefully pointed out the advantages and disadvantages of most involved methods** in our four findings. Nevertheless, we would also like to note that explaining each of these methods' advantages can be very difficult and will require a large amount of experiments to rigorously verify the reasons, which goes beyond the main focus and page limit of this paper. As suggested by the reviewer, we will add an analysis about why each method can exhibit such advantages/disadvantages, together with a detailed explanation and empirical evaluation, in the appendix of our next version. We hope this helps address your concern.
>
> ---
>
> >  **Reviewer**: For a benchmark study of fair graph learning, how do you choose the datasets for testing? What is the group density in the graphs? Are the groups balanced or not? These details could be clarified to justify the diverse choice of datasets for evaluation.
>
> **Authors**: We thank the reviewer for explaining this concern. We typically chose datasets that are most popular and widely used in the area of graph learning and applicable for study on algorithmic fairness (e.g., sensitive attributes should be included to study group fairness). These datasets are widely acknowledged by existing works as representative test base to reflect real-world decision-making scenarios. Also, we have involved two new datasets extracted from the famous AMiner platform to enrich the current choices of existing studies. As suggested, we **present a detailed collection of the statistics of the datasets** involved, and we will expand the dataset description to explicitly include more details about the adopted dataset in the appendix in our next version. We hope this helps address your concern.
>
> | Dataset    | Nodes  | Edges  | Density  | Sens 0 | Sens 0 Ratio | Sens 1 | Sens 1 Ratio |
> | ---------- | ------ | ------ | -------- | ------ | ------------ | ------ | ------------ |
> | german     | 1000   | 22242  | 0.044529 | 690    | 0.690        | 310    | 0.310        |
> | recidivism | 18876  | 321308 | 0.001804 | 9317   | 0.494        | 9559   | 0.506        |
> | credit     | 30000  | 152377 | 0.000339 | 27315  | 0.910        | 2685   | 0.089        |
> | pokec_z    | 67796  | 651856 | 0.000284 | 43962  | 0.648        | 23834  | 0.352        |
> | pokec_n    | 66569  | 550331 | 0.000248 | 47338  | 0.711        | 19231  | 0.289        |
> | AMiner-S   | 39424  | 72172  | 0.000093 | 15893  | 0.403        | 23531  | 0.597        |
> | AMiner-L   | 129726 | 655902 | 0.000078 | 52579  | 0.405        | 77147  | 0.595        |
>
> ---
>
> >  **Reviewer**: What are the connections and the comparison within different group fairness metrics, within different individual fairness metrics, and between them? Can you provide more explanations in Section 4.3?
>
> **Authors**: We thank the reviewer for this insightful comment. We agree that exploring the interplay between fairness metrics provides a deeper understanding. For group fairness metrics, such as $\Delta_{SP}$ and $\Delta_{EO}$, the connection lies in their focus on subgroup-based equity. Different from $\Delta_{SP}$ generally focusing on all positive outcomes, $\Delta_{EO}$ emphasizes positive outcomes conditioned on ground truth labels. Individual fairness metrics like $B_{Lipschitz}$ and NDCG@$k$ relate through their shared goal of preserving similar output for similar input. However, $B_{Lipschitz}$ focuses on the absolute differences while NDCG@$k$ emphasizes on the relative differences. Between these categories, a trade-off often emerges due to their differing granularities. We will include this analysis in Section 4.3 to provide a comprehensive comparison. We hope this helps address your concern.

---

> ### Author Response · Authors · 2024-11-23
> **Authors’ Response 2/2**
>
> ---
>
> >  **Reviewer**: Although the authors studied efficiency in Section 4.4, it is not well discussed as a factor in their practical guidance.
>
> **Authors**: We thank the reviewer for pointing out this limitation. We agree with the reviewer that while efficiency is presented in Section 4.4, its practical implications on method selection can be further elaborated. For instance, methods like FairWalk and CrossWalk offer a balance of fairness and computational cost, suitable for large-scale graphs. In contrast, GNN-based methods like FairGNN, though more computationally intensive, excel in versatility across fairness metrics. We will integrate these considerations into the practitioner guide in Section 5. We hope this helps address your concern.
>
> ---
>
> >  **Reviewer**: In all, despite the extensive experiments, more in-depth analysis could be added instead of simply presenting the results. Please also see the minor weaknesses in the questions.
>
> **Authors**: We thank the reviewer for this suggestion. To enhance the depth of analysis, we will revise Section 4 to include detailed discussions of why certain methods outperform others under specific fairness metrics and datasets. Additionally, we will address the minor weaknesses in our responses below.
>
> ---
>
> >  **Reviewer**: I did not find the newly constructed datasets AMiner-S and AMinder-L in the supplemental materials, which I think may count for part of the contributions of this work. Am I missing something?
>
> **Authors**: We thank the reviewer for highlighting this issue. We note that both AMiner-S and AMinder-L are large and thus are not suitable to be directly uploaded as a part of supplementary material or any online repository. We will make both new datasets publically available through online drives (e.g., Google Drive or AWS) upon acceptance. We hope this helps address your concern.
>
> ---
>
> >  **Reviewer**: In lines 334-335 you mentioned "neither DeepWalk nor GNN yields top-ranked performance under utility". Where does this statement come from? DeepWalk achieves an even worse ranking in utility than fairness.
>
> **Authors**: We thank the reviewer for mentioning this concern. We would like to clarify that **a lower ranking indicates a better performance** in Figure 2. Therefore, by mentioning "neither DeepWalk nor GNN yields top-ranked performance under utility", we would like to emphasize that fairness optimization does not always compromise utility, since these models without any fairness consideration do not bring the best performance on utility. Instead, FairVGNN enjoys the lowest average ranking, which indicates the best utility in general. We hope this helps address your concern.
>
> ---
>
> >  **Reviewer**: In Figure 3, what do dots of different colors stand for?
>
> **Authors**: We thank the reviewer for bringing up this question. The dots in different colors in Figure 3 represent various methods evaluated during the hyperparameter tuning process. Specifically, **different hyperparameters will lead to different balances** between AUC-ROC (utility level) and $\Delta_{EO}$ (fairness level). Among those cases, we can observe a clear frontier stretching between the two perspectives, indicating the best balance we can achieve. We will update the figure caption to explicitly clarify this point. We hope this helps address your concern.
>
> ---
>
> We thank you again for your valuable feedback on our work. With our further clarification, we believe that we have **responded to and addressed all your concerns with our point-to-point responses** — in light of this, **we hope you consider raising your score**.  We again sincerely appreciate the time and efforts you've dedicated to reviewing and providing invaluable feedback to enhance the quality of our paper.

---

> > ### Comment · Reviewer_NAeS · 2024-11-26
> > **Thank you for the rebuttal**
> >
> > I appreciate the clarification from the authors. However, the lack of in-depth analysis makes it hard for me to raise the score.

---

> ### Author Response · Authors · 2024-11-25
> **A Kind Reminder**
>
> Dear Reviewer NAeS,
>
> Thank you for your valuable feedback on our work. We have prepared a thorough response to address your concerns. We believe that we have responded to and addressed all your concerns with our responses — in light of this, **we hope you consider raising your score**.
>
> Notably, given that we are approaching the deadline for the rebuttal phase, we hope we can receive your feedback soon. We thank you again for your efforts and suggestions!

---

> ### Author Response · Authors · 2024-11-27
>
> We sincerely thank the reviewer for the feedback. We believe that we have responded to each of the concerns very carefully with comprehensive analysis. We kindly request the reviewer to specify which response you would like us to elaborate on so that we can provide the mentioned "in-depth analysis".
>
> We will be more than glad to proactively follow up on any concerns the reviewer may want to follow up with, and we sincerely thank you again for your feedback. We look forward to your further feedback on our submission!

---

### Meta-Review · Area_Chair_a7pz · 2024-12-17

**Metareview:**

This paper presents a comprehensive benchmark for evaluating fairness-aware graph learning methods. The authors propose a systematic evaluation protocol and conduct extensive experiments on seven real-world datasets using ten representative fairness-aware graph learning methods.

Reviewers appreciate the importance of benchmarking fairness-aware graph learning methods and acknowledge the well-written and organized nature of the paper. The proposed benchmark is seen as a valuable contribution to the field, offering insights into the strengths and weaknesses of different methods.

However, reviewers also identify some areas for improvement:

- In-depth Analysis: While the paper provides experimental results, a more in-depth analysis of why different methods exhibit different fairness-utility trade-offs is needed. This would enhance the understanding of the underlying mechanisms and provide more actionable insights for practitioners.
- Baseline with Random Attributes: Including experiments with GNN baselines that use random node attributes would help assess the impact of attribute information on fairness and provide a stronger baseline for comparison.
- Comparison with State-of-the-art: The benchmark could be strengthened by including comparisons with additional state-of-the-art fairness-aware graph learning methods, ensuring it captures the latest advancements in the field.

Recommendation:

While the paper presents a valuable benchmark for fairness-aware graph learning, the reviewers believe it could be strengthened by addressing the aforementioned weaknesses. I recommend rejecting the paper in its current form, but encourage the authors to revise and resubmit.

**Additional Comments On Reviewer Discussion:**

The submission has been strengthen through the review process, unfortunately it is still below the acceptance bar.

---

### Decision · Program_Chairs · 2025-01-22

Reject